# A viral genome packaging ring-ATPase is a flexibly coordinated pentamer

Li Dai [1,8], Digvijay Singh[2,3,6,8], Suoang Lu[4], Vishal I. Kottadiel [1], Reza Vafabakhsh [4,7],
Marthandan Mahalingam[1], Yann R. Chemla [3,4,5 ✉], Taekjip Ha [2,5 ✉] & Venigalla B. Rao [1 ✉]

Multi-subunit ring-ATPases carry out a myriad of biological functions, including genome packaging in viruses. Though the basic structures and functions of these motors have been well-established, the mechanisms of ATPase firing and motor coordination are poorly understood. Here, using single-molecule fluorescence, we determine that the active bacteriophage T4 DNA packaging motor consists of five subunits of gp17. By systematically doping motors with an ATPase-defective subunit and selecting single motors containing a precise number of active or inactive subunits, we find that the packaging motor can tolerate an inactive subunit. However, motors containing one or more inactive subunits exhibit fewer DNA engagements, a higher failure rate in encapsidation, reduced packaging velocity, and increased pausing. These findings suggest a DNA packaging model in which the motor, by re-adjusting its grip on DNA, can skip an inactive subunit and resume DNA translocation, suggesting that strict coordination amongst motor subunits of packaging motors is not crucial for function.

[1] Bacteriophage Medical Research Center, Department of Biology, The Catholic University of America, 620 Michigan Avenue, N.E., Washington, DC 20064, USA. [2] Howard Hughes Medical Institute, Department of Biophysics and Biophysical Chemistry, Johns Hopkins University School of Medicine, Baltimore, MD 21205, USA. [3] Center for Biophysics and Quantitative Biology, University of Illinois at Urbana-Champaign, Urbana, IL 61801, USA. [4] Department of Physics, University of Illinois at Urbana-Champaign, Urbana, IL 61801, USA. [5] Center for the Physics of Living Cells, University of Illinois at Urbana-Champaign, Urbana, IL 61801, USA. [6] Present address: Division of Biological Sciences, University of California, San Diego, La Jolla, CA 93093, USA. [7] Present address: Department of Molecular Biosciences, Northwestern University, Evanston, IL 60208, USA. [8] These authors contributed equally: Li Dai, Digvijay Singh. ✉email: ychemla@illinois.edu; tjha@jhu.edu; rao@cua.edu

Double-stranded DNA-containing bacteriophages are the most widely and abundantly distributed "organisms" on Earth[1,2]. Bacteriophage (phage) T4, a phage with a contractile tail that infects *Escherichia coli*, belongs to *Myoviridae* family and is one of the most well-characterized viruses[3]. It has served as an extraordinary model to study virus assembly and elucidate basic molecular biology mechanisms. T4 has also emerged as a useful platform for biomedical applications, including vaccine delivery and gene therapy[4].

Molecular motors belonging to the additional strand conserved E (glutamate) superfamily are responsible for a myriad of cellular activities, including DNA replication involving DNA helicases, transcription regulation by transcription termination factors such as Rho, protein degradation by chaperones and proteasomes, and cargo transport by dyneins[5]. These molecular machines are ring-NTPases powered by energy from NTP hydrolysis, and many translocate their substrates through the ring's central channel. The packaging motors, such as the large "terminase" gp17 of phage T4, share this common characteristic of being homomeric ring-ATPases that utilize ATP energy to encapsidate viral genomes[6,7]. Thus, they are a useful model for studying conserved biological mechanisms of ring-NTPases such as motor coupling and coordination of chemical events for processive mechanical movement[8]. Since the basic packaging mechanism is well-conserved in eukaryotic viruses such as herpes viruses, genome packaging also presents a special target for antiviral drug discovery[9].

Phages employ powerful molecular motors to translocate DNA into a pre-assembled capsid shell[6,7]. They generate extraordinary forces, as high as 80–100 pN, >25 times that of myosin, to overcome the mechanical and repulsive electrostatic forces from the DNA condensed inside the capsid[10]. In T4, a ring of gp17 molecules assemble on an empty prohead at the unique portal vertex of the icosahedral capsid[11] and encapsidate ~171 kb (headful) of newly synthesized concatemeric genomic DNA[11]. Genetic and biochemical analyses have established that gp17 possesses ATPase, nuclease, and translocase activities essential for cutting the concatemeric genome and translocating a unit length DNA genome into the virus capsid[12–14]. Packaging at a rate of up to ~2000 bp/s and generating a power density of ~5000 kW/m³, about twice that of a car engine, the T4 motor is the fastest and the most powerful packaging motor reported to date[15].

The atomic structure of phage T4 packaging motor protein gp17[11] and of the analogous proteins from other phages and herpes viruses[16–21] have been determined. These display a common architecture containing two major domains connected by a linker or hinge domain. The N-terminal ATPase domain powers DNA translocation whereas the C-terminal nuclease domain makes cuts in the concatemeric DNA at the time of packaging initiation and termination[11]. The C-terminal domain is also required for DNA binding and translocation functions[6,7]. The motor protein of phage ϕ29 also retains this common architecture[22], despite lacking nuclease activity because it packages a unit-length genome and does not require initiation and termination cuts.

Extensive structural and functional studies of packaging motors support a common mechanism, and several models have been proposed[11,22–26]. However, many aspects of the packaging mechanism remain unresolved or are controversial. Cryo-electron microscopic (cryo-EM) reconstructions have determined a pentameric stoichiometry of capsid-assembled motors in T4 and ϕ29 phages[11,21]. However, the motor in the case of T4 was in a passive state, not associated with DNA. In the case of ϕ29, counting of fluorescently labeled packaging RNAs associated with the ϕ29 packaging motor, in both passive and active states, gave a distribution consistent with a hexameric stoichiometry[27]. Furthermore, little is known about how the motors engage with the DNA

substrate, the dynamics of DNA engagements at the time of packaging initiation, and how the firing of motor ATPases is coordinated. Based on a structural model of the T4 motor, it was proposed that each subunit takes turns and sequentially hydrolyze ATP to move ~7 Å (or ~2 bp) of the DNA into the capsid[11]. Studies of ϕ29 and λ phage in which packaging motors were doped with inactive ATPases suggest a strict coordination mechanism[28–30]. However, these measurements were made using either ensembles of motors with heterogeneous compositions or were done under conditions where the number of inactive subunits could not be directly ascertained.

Here we report measurements using single-molecule total internal reflection fluorescence (TIRF) microscopy and optical traps that allow real-time observation of DNA engagements and packaging by individual T4 motors. The fluorescence experimental design allows to precisely count the number of active/inactive motor subunits in each motor and observe packaging by individual motors in real time. These studies reveal several insights. (1) We find that the stoichiometry of the actively packaging phage T4 motor is five. (2) Though motors repeatedly engage DNA, many of these engagements are unsuccessful, i.e., not proceeding to DNA encapsidation. (3) Notably, motors containing one or more inactive subunits are capable of packaging DNA, although the presence of even one inactive subunit reduces motor's ability to engage with DNA and complete packaging. (4) All the motors (including the ones with inactive subunits) alternate between an active DNA engaging state and a quiescent state where no DNA engagements occur, but the motors with inactive subunits spend longer time in the quiescent state. (5) Motors containing at least one inactive subunit exhibit a ~1.5-fold decrease in packaging velocity and a ~2-fold increase in pausing during DNA translocation compared to wild-type (WT) motors. These findings indicate that the T4 motor's homomeric subunits are not strictly coordinated and can tolerate inactive subunits, suggesting a revised packaging mechanism and a refined understanding of T4 motor's dynamic engagements with DNA.

## Results

**The actively packaging T4 motor is a pentamer**. We used single-molecule TIRF to count the number of gp17 subunits present in an actively packaging T4 motor, i.e., the motor's stoichiometry. A functional T4 motor was re-constituted by mixing empty T4 heads (capsids) and gp17 in the presence of the poorly hydrolyzed analog ATP-γS. A 120-bp unlabeled DNA was also added to allow efficient assembly of packaging-competent motors, which requires both ATP-γS and DNA. The assembled motors were then attached to a polyethylene glycol (PEG) passivated surface via anti-T4 polyclonal antibody (Fig. 1a and Supplementary Fig. 2a; see "Methods")[31]. The motor comprised of active gp17 subunits into which a fluorescent label was introduced by fusing a 20-kDa SNAP-tag at the N-terminus and labeling it with the SNAP-Surface® 549 fluorescent substrate (hereafter referred to as "Cy3" because it has similar fluorescent properties as Cy3) (Supplementary Fig. 1a–c; see Methods). Each packaging motor was imaged as a single fluorescent spot consisting of *m* Cy3 labels on *n* subunits, with *m* depending on labeling efficiency and photoactivity. Each motor's packaging activity was monitored from the fluorescence of encapsidated short, 45-bp Cy5-labeled double-stranded DNA molecules (Fig. 1a). Correspondingly to each Cy3 spot, a colocalized Cy5 spot indicated packaged Cy5-DNA, certifying the motor's activity (Fig. 1b inset). Using a bulk DNA packaging assay, we verified that Cy3 SNAP-tagged motors were nearly as active as the WT motors (Supplementary Fig. 1c). The appearance of Cy3 and Cy5 signals required the presence of all the components of an immobilized packaging motor,

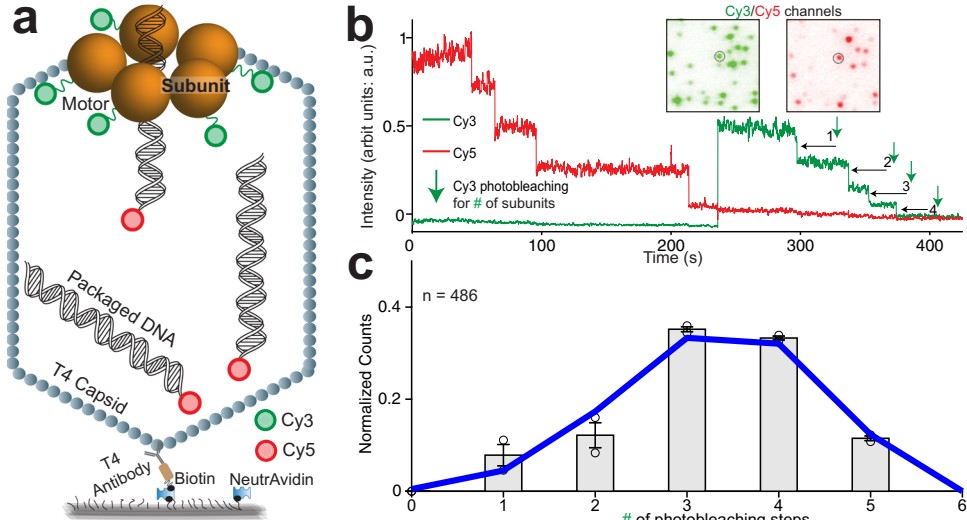

**Fig. 1 Single-molecule fluorescence assay determines a pentameric stoichiometry for the actively packaging T4 motor. a** Schematic of the assay. Homomeric subunits constituting the motor are labeled with Cy3 and 45-bp dsDNA molecules are labeled with Cy5. **b** Representative fluorescence time trajectories showing Cy3 and Cy5 signals, which indicate T4 motor and packaged DNA, respectively. The indicated Cy3 photobleaching steps are used to quantify the stoichiometry of the T4 motor (# of subunits forming the motor). The presence of the Cy5 signal indicates that the motor is capable of packaging DNA. The number of Cy5 photobleaching steps, indicating number of packaged DNA, can range from 1 to many. Inset: CCD images of the Cy3 and Cy5 channels from which the representative trajectories were selected. **c** The distribution of Cy3 photobleaching steps pooled from separate experiments and a binomial fit (blue curve) to the distribution with $n = 5$ subunits (stoichiometry of the motor) and $p = 0.66$ (Cy3 labeling efficiency). Error bars represent std. The number of trajectories used is indicated by $n = \#$.

validating the sensitivity and accuracy of our assay (Supplementary Fig. 2b).

To quantify stoichiometry of the motor, we first added 1 mM ATP and 2.5 nM Cy5-DNA into the imaging chamber and allowed packaging to proceed on the surface for ~10 min. Then packaging was terminated by adding DNase I to digest any DNA that was not encapsidated. Next, Cy3–Cy5 were imaged until complete photobleaching to visualize the intensity of Cy3 and Cy5 spots as a function of time in the form of single-molecule trajectories (Fig. 1b, c). For each of the trajectories exhibiting both Cy3 and Cy5 signals, indicating an active motor, the photobleaching steps of Cy3 intensity were counted to determine the number of Cy3-labeled gp17 subunits per motor (Fig. 1b). These Cy3 counts from multiple trajectories were pooled together into a distribution (Fig. 1c). The binomial distribution was used to describe the experimental distribution, with two fitting parameters: the number of subunits in the motor $n$ and the Cy3 labeling efficiency $p$ of the gp17. The best fit of the distribution was with $n = 5$ and $p = 66\%$, also in agreement with independent measurements of labeling efficiency (see "Methods"; Fig. 1c, solid blue line and Supplementary Fig. 1d). In contrast, the binomial distributions with $n = 4$ or $n = 6$ were improbable. Many trajectories with counts of 5 were observed invalidating the $n = 4$ and no trajectories with a count of 6 photobleaching steps (among 486 trajectories analyzed) were found invalidating $n = 6$ (Supplementary Fig. 3a). Thus, this distribution and their fits indicate the stoichiometry of the actively packaging T4 motor to be five gp17s.

**The T4 motor can tolerate inactive subunits**. Next, we investigated the coordination of the five gp17 subunits by analyzing the packaging behavior of motors containing inactive subunits. For the inactive subunit, we selected a gp17 mutant in the Walker A P-loop, Q163A[32,33], which is impaired in ATP hydrolysis (Supplementary Fig. 1e) and shows no detectable levels of DNA packaging activity in ensemble assays (Supplementary Fig. 1c).

Previous crystal structures[34] show that the Q163 residue is not in direct contact with the bound ATP, suggesting that the Q163A mutant is capable of binding ATP but deficient in a post-ATP binding step. The Q163A mutant binds to T4 heads with the same affinity as WT gp17 and can also assemble into a pentamer (Supplementary Fig. 3b–g). We doped the packaging reaction with a varying fraction of inactive subunits. Since gp17 exists as a monomer in solution and assembles as a pentamer on the portal vertex, doping led to an ensemble of hetero-pentameric motors containing between 0 and 5 inactive subunits. Figure 2a, b show that the packaging activity in bulk, as determined from the amount of DNA protected from DNase I digestion by encapsidation (see "Methods"), decreases as the fraction of inactive subunits increases (Fig. 2b). A similar trend was also observed at the single-molecule level, where the number of Cy5 spots (each indicative of 1 or more packaged DNAs) decreased with an increasing fraction of inactive subunits (Fig. 2c–e).

Based purely on the statistical probabilities, we determined the relative populations of motors containing 0–5 inactive subunits. These populations were used to predict the ensemble activity at varying levels of coordination (see "Methods"). In a strictly coordinated motor, firing by one subunit depends on the firing of an adjacent subunit[21,28,29]. Therefore, the presence of one inactive subunit should stall the motor, and only those motors with 5 active subunits would exhibit packaging (Fig. 2e, solid black line). Conversely, in a completely uncoordinated motor, the packaging activity would continue with even 1 active subunit. As a result, the measured packaging activity should depend linearly on the fraction of active subunits (Fig. 2e, solid gray line). At intermediate coordination levels, a minimum of 2–4 active subunits would be required for packaging, with a motor containing 4 active subunits exhibiting twice the activity of a motor with 2 active subunits (Fig. 2e, dashed and dotted lines). Comparing the experimental data to these models shows that the packaging activity exceeds that predicted for a strictly coordinated motor but is also lower than that predicted for a completely uncoordinated motor (Fig. 2e). Notably, this comparison suggests

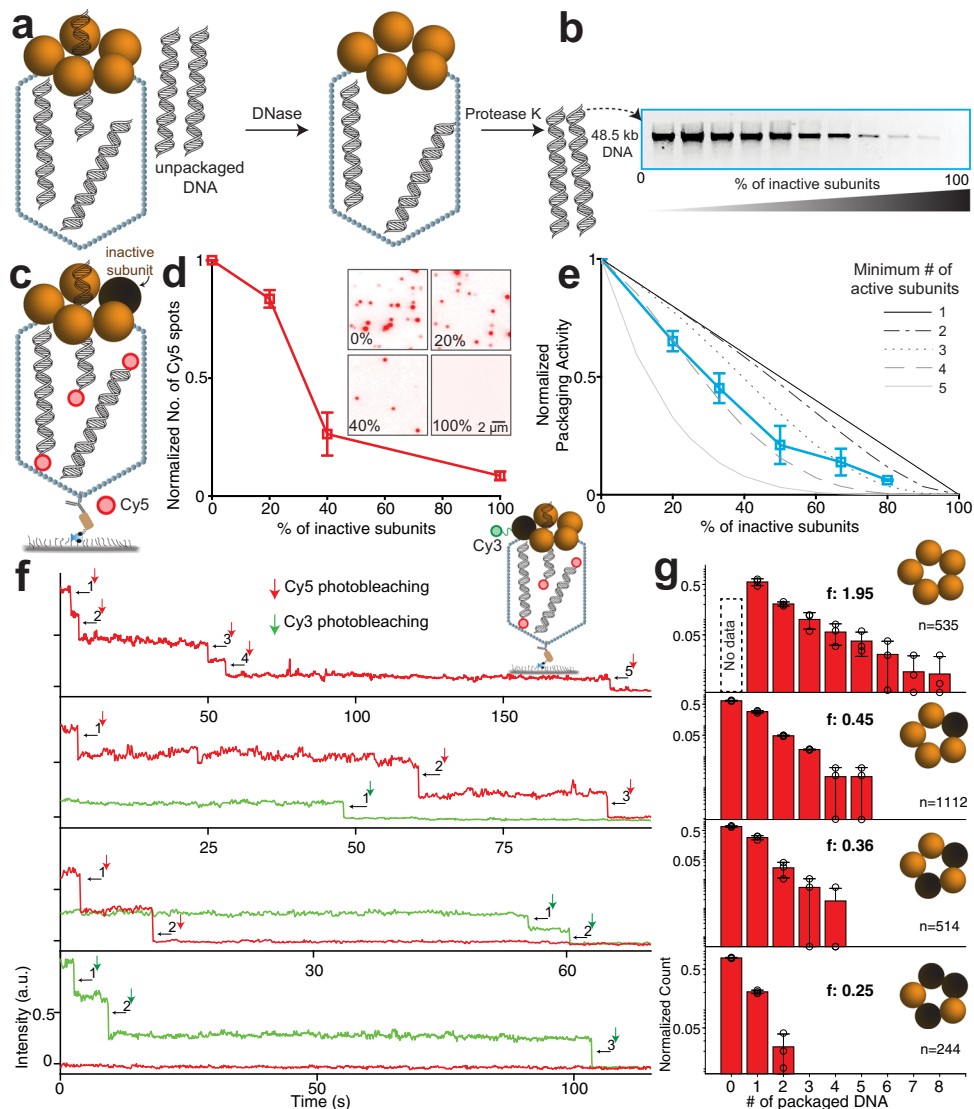

**Fig. 2 Motors containing inactive subunits exhibit impaired, but not abolished, activity. a** Schematic of in vitro bulk packaging assay. After allowing the packaging to proceed, the unpackaged DNA was digested by DNase I and the packaged DNA was retrieved from the capsids by digesting the capsid and DNase I with proteinase K. **b** Agarose gel electrophoresis of DNA packaged by motors constituted with increasing numbers of inactive subunits. The WT gp17 and the inactive Q143A mutant gp17 (inactive subunit) were pre-mixed at different ratios and added to the reaction mixtures containing capsids, ATP, and linearized plasmid DNA. The uncut gel image is shown in Supplementary Fig. 14. **c** Schematic of single-molecule fluorescence assay. The assay is similar to the one in Fig. 1a but performed with increasing percentage of inactive subunits (black sphere). **d** No. of Cy5 spots with increasing percentage of inactive subunits. In the inset are representative CCD images of the Cy5 channel, showing the number of Cy5 spots (each containing 1 to n # of packaged DNA) indicate the packaging efficiency with increasing percentage of inactive subunits. **e** DNA packaging activity quantified from the agarose gel in **b** (cyan squares). Also shown are predicted activities where the minimum number of active subunits required for packaging is 1 (no coordination; black) to 5 (strict coordination; gray)). **f** Representative single-molecule trajectories of Cy5 photobleaching steps (red), indicating the number of DNA molecules packaged inside the capsid, and Cy3 photobleaching steps (green), indicating the number of inactive subunits in the motor. Inset: Schematic of the assay, similar to the one shown in **c**, with inactive subunits (black sphere) labeled with Cy3. **g** Distribution of the counts of packaged DNA molecules from motors reporting 0–3 Cy3 photobleaching events (indicative of the count of inactive subunits). The average number of packaged DNA is indicated by *f*. The number of trajectories used is indicated by n = #.

that the presence of at least one inactive subunit does not abolish motor activity and that the motor is not strictly coordinated.

To further probe into the mechanism of packaging for motors containing inactive gp17 subunits, we selected single inactive-subunit doped (ISD) motors containing a precise number of inactive subunits and analyzed their packaging behavior. We used a single-molecule fluorescence assay, similar to the one above, but with Cy3 labeling of only the inactive Q163A gp17 subunit, which was mixed with unlabeled WT gp17 to assemble motors with varying numbers of inactive subunits. The number of Cy3

photobleaching steps indicated the number of inactive subunits in each motor (# Cy3), and the Cy5 photobleaching steps indicated the number of packaged DNA molecules (# packaged DNA). Packaging was initiated by flowing Cy5-DNA and ATP into the chamber containing the surface-immobilized motors and terminated by addition of DNase I after ~10 min. Strikingly, many spots colocalized both Cy3 and Cy5 fluorescence, providing direct confirmation that motors containing the inactive subunit(s) were able to package Cy5-DNA (Fig. 2f, g; 2405 trajectories were analyzed). Due to <100% labeling efficiency and photoactivity, a

small fraction of the photobleaching steps in the analysis shown in Figs. 2 and 3 may represent more inactive subunits than the ones represented by photobleaching steps. No. of.

The number of packaged DNA molecules for varying number of Cy3 inactive subunits were pooled from multiple trajectories to generate a distribution of the number of packaged DNA as a function of the number of Cy3-labeled monomers (Fig. 2g). While the ISD motors containing 1 or 2 inactive subunits could package multiple Cy5-DNA molecules (Fig. 2f, g), packaging overall was impaired when compared to WT motors. Although we could not determine how often WT motors (which are unlabeled) failed to package DNA (# packaged DNA = 0) since such events generate no Cy3 fluorescence signal, the differences in the distributions of number of packaged DNA molecules for WT and ISD motors was significant. While the WT motors could be found packaging as many as ≥8 DNA molecules within the ~10 min allotted, this number rarely exceeded 2 or 3 for ISD motors (Fig. 2g). Strikingly, the number of packaged DNA was not proportional to the number of inactive subunits, indicating that packaging is impaired by the presence of an inactive subunit but not necessarily proportional to the number of inactive subunits (Fig. 2g).

**Motors with inactive subunits show a reduced propensity to engage DNA and a reduced packaging efficiency**. To probe the mechanism by which the motors engage with DNA and initiate packaging, we modified the single-molecule assay described in Fig. 2c to monitor the Cy5 fluorescence intensity in real time, immediately after the addition of ATP and Cy5-DNA (Fig. 3a and Supplementary Figs. 4–6; see "Methods"). The Cy5 intensity trajectories for single packaging motors showed sharp upward steps, indicating real-time DNA engagements by the motor, as reported previously[31]. The sharp downward steps in the Cy5 signal indicated either dissociation of the DNA or Cy5 photobleaching. At the end of each trajectory, we counted the number of inactive subunits in each motor from Cy3 photobleaching steps (Fig. 3a; 2888 trajectories were analyzed).

The dwell times between the Cy5 upward steps represent the time for DNA to engage the motor[31] (Fig. 3a). For motors with 0, 1, and 2 inactive subunits, we collected individual dwell times from multiple trajectories and calculated the average lifetime ($\tau_{avg}$). $\tau_{avg}$ was ~12 s for WT motors and rose to ~40 s for motors with 1 or 2 inactive subunits, highlighting a reduced propensity (or ability) of the ISD motors to engage the DNA (Fig. 3b). Consistent with previous studies[31], the distribution of these dwell times was best described by a double exponential function, with two lifetimes, $\tau_s$ and $\tau_l$ ($\gg \tau_s$) ($\tau_{avg}$ being the amplitude-weighted average of $\tau_l$ and $\tau_s$; Supplementary Fig. 7). It was previously shown that the T4 motor alternates between an "DNA engaging" state competent to engage and package DNA in rapid bursts and a long-lived "quiescent" state that does not engage DNA at all[31]. $\tau_s$ corresponds to the time for DNA to bind to the motor in its active state, whereas $\tau_l$ represents the lifetime of the quiescent state. $\tau_s$ did not change appreciably with inactive subunits (Fig. 3b), indicating that the time for DNA engagement in the active state was unaffected. In contrast, $\tau_l$ more than tripled in the presence of inactive subunits (Fig. 3b), indicating that the ISD motors had a longer-lived quiescent state.

We also estimated the lifetimes of the DNA engagements from the dwell time between each upward step and the subsequent downward step in Cy5 fluorescence intensity (see "Methods"). We observed different characteristic types of DNA engagements. Many DNA engagements were transient, lasting <2 s (Fig. 3a, purple), much shorter than the average photobleaching lifetime of Cy5 under the imaging conditions (expected to be ~40 s). We

interpreted the downward steps in fluorescence during these transient engagements as dissociations of Cy5-DNA from the motor, i.e., DNA engagements during which packaging failed. In contrast, we also observed long-lived engagements lasting >10 s (Fig. 3a, cyan). The downward steps ending these engagements were consistent with Cy5 photobleaching, and such events likely represent successful packaging events with completed encapsidation of 45 bp long DNA. Lastly, we detected engagements of intermediate lifetime, "semi-transient" engagements that lasted between 2 and 10 s (Fig. 3a, gray). These could plausibly represent engagements in which packaging was initiated but failed before complete encapsidation occurred. Distributions of DNA engagement lifetimes were best fit by a three-exponential model, validating our classification of DNA engagements into three characteristic types (Supplementary Fig. 8). The frequency of all three types of engagements is plotted for motors with 0–2 inactive subunits (Fig. 3c). For all motors, the frequency of transient and semi-transient engagements was significantly higher than those of long-lived engagements. This observation signifies that all motors, irrespective of the number of inactive subunits, tend to fail frequently in attempts to package DNA (from an end). In addition, there was a significant difference in the frequency of long-lived DNA engagements between the WT and ISD motors (Fig. 3c). The fraction of DNA engagements that were long-lived was higher for WT motors than for ISD motors (Fig. 3d), indicating an impaired mechanism of successful packaging in the ISD motors.

We estimated the total number of DNA molecules packaged by a motor from the cumulative number of long-lived engagements until the endpoint of the trajectory. These numbers, from multiple trajectories, were pooled together into a distribution for motors with 0, 1, and 2 inactive subunits (Fig. 3e). These distributions show that, in the first 4 min, the WT motor can package an average of 0.81 Cy5-DNA molecules when compared to 0.27 and 0.32 for motors with 1 and 2 inactive subunits, respectively. Markedly, there were multiple instances of WT motor being able to package as many as 5–7 DNAs. However, this number rarely exceeded two packaged DNA molecules for motors containing inactive subunits.

We also observed that the engagements of the same type tended to cluster together in the time trajectories. We estimated the probabilities of each type of engagement following the engagement of each particular type. The probability of a transient engagement following another transient engagement was quite high (~0.9) (Supplementary Fig. 9s). The pattern was similar for the long-lived and semi-transient engagements, confirming that motors (in DNA engaging state) undergo persistent phases of repeated successful DNA engagements or unsuccessful DNA engagements. Detailed comparisons between the WT motor with and without the SNAP tag, further confirmed that SNAP tag did not affect the motor function (Supplementary Fig. 10).

**Motors doped with inactive subunits show frequent pausing and reduced packaging velocity**. The above measurements provide a picture for how the T4 motor engages DNA to initiate packaging. To probe the steps following initiation, we employed a dual-trap optical tweezers assay to monitor individual motors packaging long (3400-bp) DNA substrates. Motors were pre-assembled on the free end of DNA molecules attached to micron-sized polystyrene beads in the presence of 1 mM of poorly hydrolyzed ATP analog, ATP-γS (see "Methods"). Due to the lack of an energy source, the motors could bind (and grip) DNA but could not package it[35]. The pre-assembled motors on beads were then flushed into the optical trap sample chamber and captured in one trap, and a tether was formed by binding a second trapped

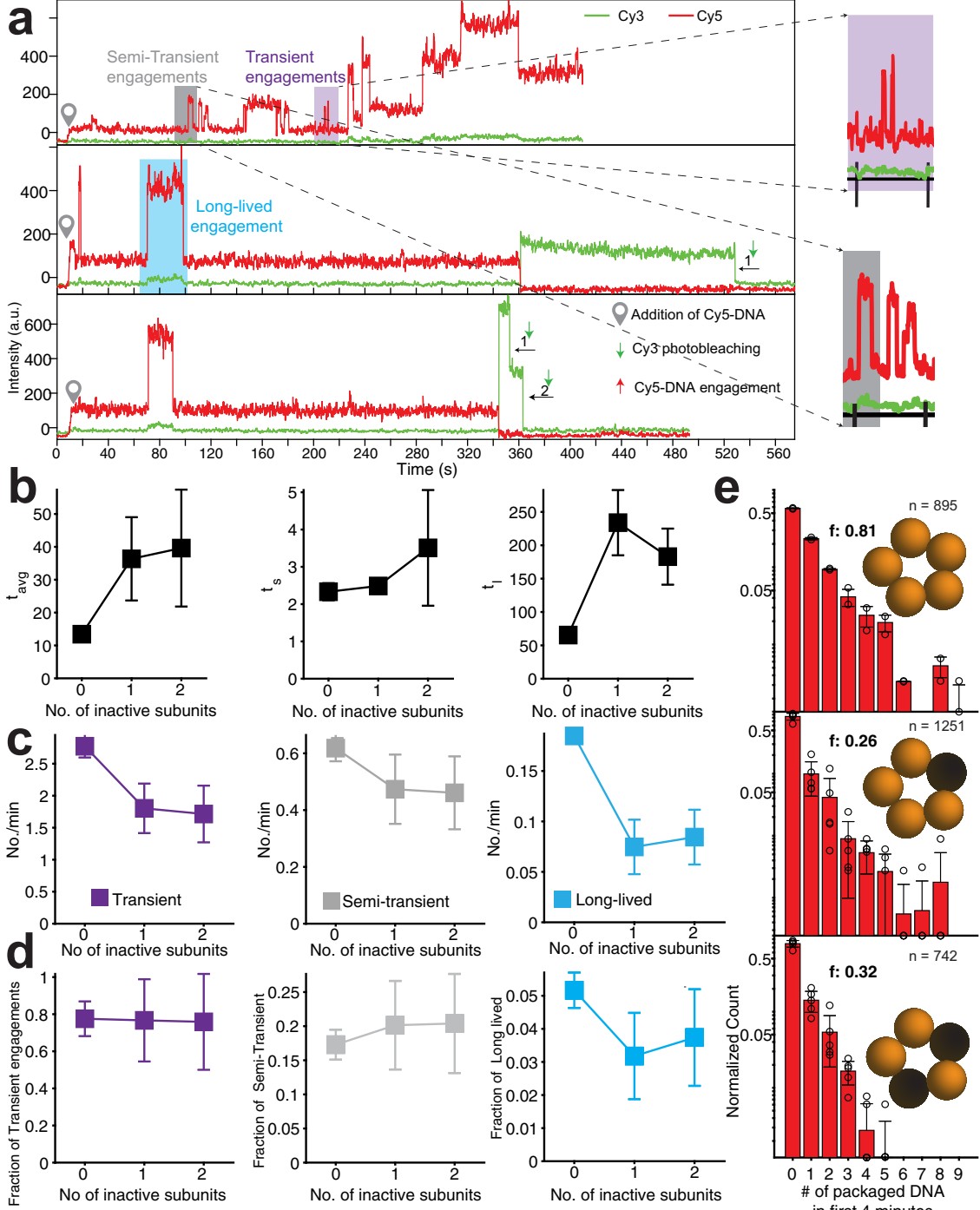

**Fig. 3 Motors with inactive subunits have less frequent and less productive engagements with DNA. a** Representative trajectories showing Cy5-DNA engaging with the motors in real-time and Cy3 photobleaching steps, indicating the number of the inactive subunits in the motor. The time point at which Cy5-DNA was introduced is shown as a pin and is manifested as a small increase in Cy5 intensity. The upward spikes in Cy5 intensity correspond to DNA engagements with the motor. Examples of three types of DNA engagements are shown: transient (purple shaded area and zoom-in), semi-transient (gray area and zoom-in), and long-lived (cyan area). **b** Dwell times between DNA engagements for motors with 0, 1, or 2 inactive subunits. The survival probability distributions for dwell times from multiple trajectories were fit to a double exponential function (Supplementary Fig. 7), as observed previously[31], yielding two lifetimes: $\tau_s$ for dwell times between DNA engagements occurring in short bursts (middle), $\tau_l$ for those occurring with long delays (right). $\tau_{avg}$ is the amplitude-weighted average of $\tau_s$ and $\tau_l$ (left). **c** Number of transient (left), semi-transient (middle), and long-lived (right) DNA engagements per minute for motors with 0, 1, or 2 inactive subunits. **d** Fraction of DNA engagements that are transient (left), semi-transient (middle), and long-lived (right) for motors with 0, 1, or 2 inactive subunits. **e** Distribution of the number of packaged DNA in the first 4 min for motors with 0, 1, or 2 inactive subunits. The average number of packaged DNA is indicated by $f$. The number of trajectories used is indicated by $n = \#$.

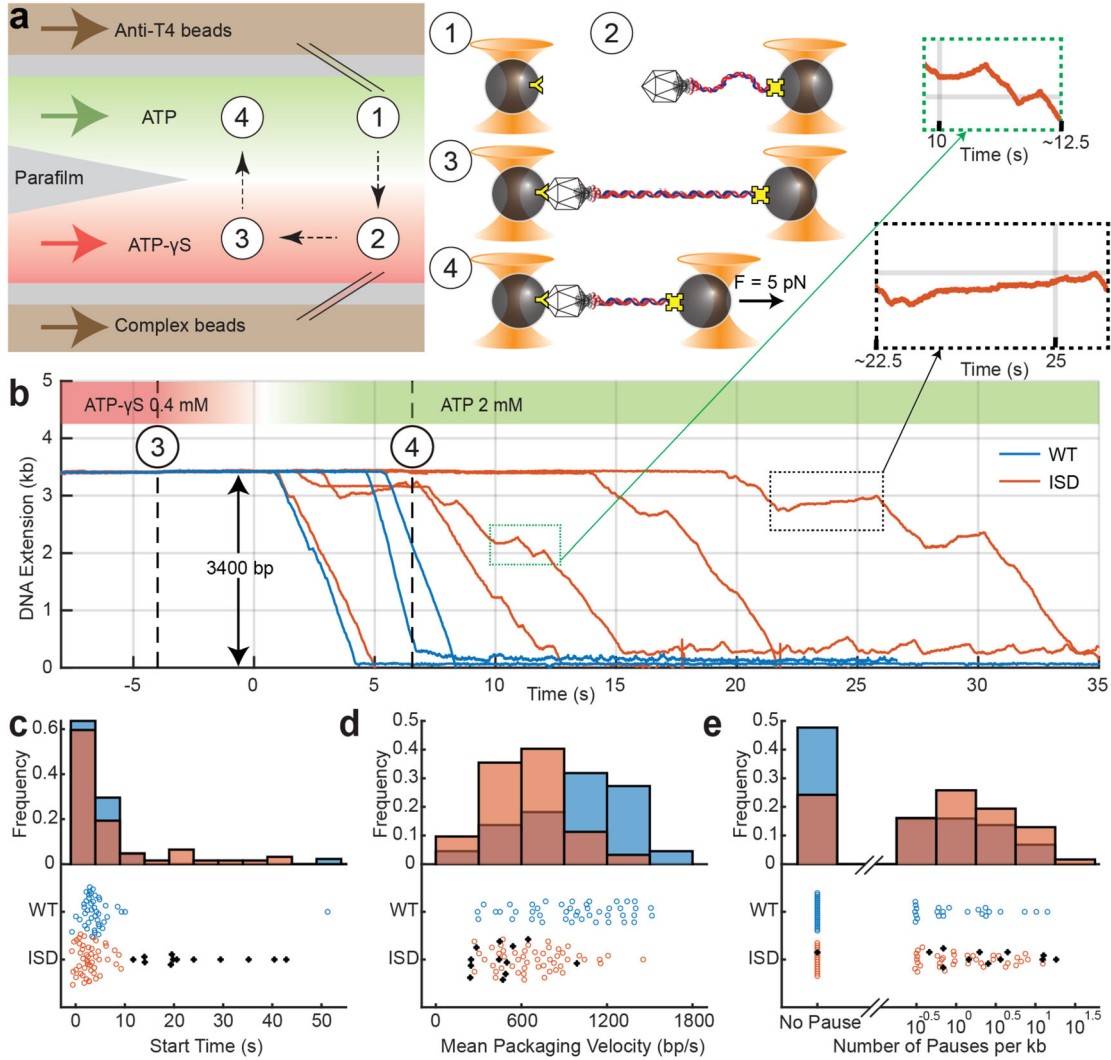

**Fig. 4 Motors containing inactive subunits exhibit slower start times, lower packaging speeds, and more pausing. a** Schematic of the optical trapping assay to measure the packaging dynamics of a single T4 motor. A multi-stream laminar flow chamber (left) is used to manipulate samples and buffer. A bead coated with anti-T4 antibodies is captured in one trap in a flow stream containing ATP (green stream; step ①). Another bead conjugated with arrested T4 capsid–gp17-DNA motors is captured in a second optical trap in the stream containing non-hydrolyzable analog ATP-γS (red stream; step ②). The motor bound to DNA in the presence of ATP-γS cannot initiate packaging due to a lack of energy source. To form a tether between the bead pair, the beads are moved close for 1 s and separated apart to detect binding of capsid to antibody (step ③). After a constant force ($F = 5$ pN) load is applied to the tether, the traps are moved into ATP and packaging is detected as the decrease of DNA extension when DNA is translocated into the capsid by the motor (step ④). **b** Extension of unpackaged DNA vs. time, under constant force load. Packaging activity is observed within seconds of entering the ATP stream. Representative packaging trajectories with WT (blue) and ISD motors (orange) are shown. **c** Distribution of the start times of WT (blue, $n = 44$) and ISD motors (orange, $n = 62$), defined as the dwell time between entry into the ATP stream and the start of packaging. **d** Distribution of mean pause-free packaging velocity derived from each packaging trace. ISD motors (orange) show a lower packaging velocity than WT motors (blue). **e** Histogram of the logarithm of pause frequency, defined as the number of pauses per kb packaged. ISD motors (orange) exhibit more pauses than WT motors (blue) on average. Motors with long start times (black stars; **c**–**e**) have a lower packaging velocity and higher pause frequency. The number of trajectories used is indicated by $n = \#$.

bead to the 1T4 capsid (Fig. 4a). This tether was stretched to a constant force of 5 pN in a channel containing 0.4 mM ATP-γS (see "Methods"). As the motor–DNA grip remained stable under applied force, we referred to these motors as "arrested" because initial engagement of the DNA by the motor was successful but translocation was stopped. The trapped and tethered bead pair was then moved from the ATP-γS channel to a channel containing 2 mM ATP (Fig. 4a) to start packaging against a constant load force. The packaging was measured from the decrease in DNA extension between the two beads, as DNA was translocated into the capsid by the motor (Fig. 4b). Different dynamic properties of individual motors were derived from these raw extension trajectories.

We investigated both WT and ISD motors. Arrested ISD motors were prepared using a 1:1 mixture of WT and Q163A gp17, in which 97% of the motors would be expected to have at least one inactive subunit (47% with 1 or 2 inactive subunits), and 3% of motors would have no inactive subunit. We selected packaging trajectories from 44 WT and 62 ISD motors for which the DNA substrates exhibited the correct length and elastic behavior (see "Methods"). We first determined the fraction of tethered motors that started packaging upon exposure to ATP in the ATP channel. We found that ~30% of WT motors started within a waiting time of 1 min in contrast to ~15% of ISD motors. Although the starting efficiency for ISD motors is lower than that

of WT motors, it is still much larger than the 3% expected if only motors containing only WT subunits can package. This observation is consistent with our bulk and single-molecule fluorescence data described above which showed that inactive subunits are tolerated for packaging. Specifically, the 50% decrease in packaging efficiency between WT and ISD motors is in good agreement with the expected fraction of complexes with 0, 1, or 2 inactive subunits.

We then compared the start times, determined by the time elapsed between crossing the interface between the ATP-γS and ATP streams (Fig. 4b; $t = 0$ s) and the first instance of packaging. The start time represents the time required for the motors to come into correct alignment with the already threaded DNA and persistently hydrolyze ATP but not the time required for DNA engagements and threading. The average start times of the arrested motors were $4.8 \pm 1.1$ s (mean ± SEM) for WT (Fig. 4b, blue) and $7.0 \pm 1.3$ s for ISD motors (Fig. 4b, orange). Significantly, while most ISD motors showed a similar start time distribution as WT motors, 20% of ISD motors restarted after >11 s, longer than all but one WT motor (Fig. 4c, black). These long start times account for the difference in means between ISD and WT motors.

Upon a successful start to packaging, the DNA extension decreased (Fig. 4b). For both WT and ISD motors, active packaging was occasionally paused, which we defined as any non-packaging interval of duration >0.1 s (see "Methods"). We determined the mean pause-free packaging velocity for each motor to assess the effect of inactive subunits. Histograms of the mean pause-free packaging velocity of WT motors (blue) and ISD motors (orange) are shown in Fig. 4d. The average velocity for ISD motors was $650 \pm 30$ bp/s (mean ± SEM), significantly lower than $960 \pm 50$ bp/s for WT motors. Mean packaging velocities of WT motors varied widely from 100 to 2000 bp/s (the standard deviation in velocity represents 35% of the mean), consistent with a previous study[15]. While ISD motors exhibited a similar variation in mean packaging velocities (40% of the mean), a smaller fraction than the WT packaged at high speeds. Differences between WT and ISD motors were thus manifested in differences in mean velocities. We could not ascribe variability in packaging velocity to the number of inactive subunits in ISD motors since WT subunits themselves account for the majority of this variability.

We also estimated pause duration and pause frequency, the latter from the number of pauses per kb packaged. On average, pauses were longer and more frequent for ISD motors ($1.5 \pm 0.6$ s, $2.3 \pm 0.5$ kb$^{-1}$) than WT motors ($0.8 \pm 0.3$ s, $1.2 \pm 0.4$ kb$^{-1}$; see Supplementary Fig. 11a and Fig. 4e). For both types of motors, we observed a negative correlation between mean packaging velocity and pause frequency (Supplementary Fig. 11b), indicating that pausing is more likely to occur when translocation is slowed. Furthermore, the 20% of ISD motors that started packaging the slowest (>15 s) also showed a lower packaging velocity ($460 \pm 60$ bp/s; Fig. 4d) and a significantly higher pause frequency ($4.9 \pm 1.7$ kb$^{-1}$) than the other 80% of ISD motors ($690 \pm 40$ bp/s and $1.7 \pm 0.4$ kb$^{-1}$, the latter close to the value for WT motors; Fig. 4e). These correlations suggest that ISD motors that are more likely to be defective in starting also show greater packaging impairment.

Lastly, we observed that pauses either consisted of plateaus, during which packaging stopped, or gradual release or "unpackaging" of DNA, which is different from "slipping" where the motor completely loses grip on DNA (Fig. 4b; enlarged insets at $t = 10$ s, 22.5 s). Both types of pauses have been previously reported in optical trap studies under low ATP conditions, which was attributed to reduced grip on DNA[35,36]. We analyzed the velocity during all pauses to determine if there was more unpackaging in

ISD vs. WT motors. The majority of ISD motors showed a similar distribution as WT motors, but ~10% showing significantly faster unpackaging events (velocity ranging from −100 to −500 bp/s), indicating motors with a poorer grip on DNA (Supplementary Fig. 11c). Furthermore, comparing these ISD motors to the other 90%, there was no significant increase in pause frequency, pause duration, or decrease of packaging velocity.

## Discussion

Our newly developed single-molecule assays allowed us to analyze the DNA packaging process through multiple stages, starting with the initial interactions between DNA and motor, engagement of DNA by the motor, translocation, and complete encapsidation. By studying individual motors with a precise number of active/inactive subunits, we could connect the above dynamic behaviors to motor stoichiometry and coordination. The determined pentameric stoichiometry of packaging-active T4 motors (Fig. 1c) is consistent with that of phages ϕ29[22] and T4 observed in cryo-EM reconstruction in which the motor was in a passive state, i.e., not engaged with DNA[11]. That these independent approaches have arrived at the same pentamer stoichiometry is significant and also in functional alignment with the 10-bp helical pitch of DNA and ~2-bp motor step-size.

We discovered, by three different assays (ensemble, single-molecule fluorescence, and optical trap) that motors with an inactive subunit are able to encapsidate DNA (Figs. 2–4). Furthermore, in single-molecule fluorescence assays where the number of inactive subunits in a motor could be counted, the data showed that two, or even three, inactive subunits were tolerated (Figs. 2g and 3e), providing a clear indication that the coordination behavior of the T4 motor is distinct from that observed in ϕ29 and λ packaging motors[21,28–30]. In the case of phage λ[29], based on statistical probabilities of packaging activity in ensemble measurements of motors doped with a Walker A mutant in the large terminase subunit, it was concluded that the motor is strictly coordinated (100% coordination). Thus, the presence of a single mutant subunit rendered the motor inactive. In the case of phage ϕ29, using a trans-arginine finger mutant in the terminase and single-molecule optical tweezers assays, it was found that one inactive subunit is tolerated[30]. However, such motors systematically pause every 10 bp in the dwell phase of each translocation cycle, resulting in a tenfold reduction in the rate of packaging. Furthermore, ϕ29 motors containing more than one defective subunit are completely inactive. The above behaviors were not observed in T4 ISD motors. In contrast, randomly occurring pauses and unpackaging events were observed (Fig. 4b), and motors containing two or even three inactive subunits are functional (Fig. 2g). However, our data also do not support a completely uncoordinated mechanism because the ISD motors manifest impaired packaging, as evident from reduced DNA engagements and fewer encapsidations (see below). Packaging of DNA is a complex process requiring carefully orchestrated steps of ATP binding, DNA engagement, ATP hydrolysis, and directional DNA movement, which have to be precisely coordinated among the motor's subunits.

In the initial stages of packaging, T4 motors alternate between a quiescent state, which is free of DNA association, and a DNA engaging state, which is competent to engage and package DNA (Fig. 5a). The inclusion of a single inactive subunit leads the motor to spend ~4× longer time in the quiescent state (Fig. 3b, middle panel). The molecular basis for the quiescent state is unclear but most likely due to impaired binding and/or hydrolysis of ATP of the inactive subunits that interferes with the transition to the DNA engaging state. Once in the engaging state, however, the rate of DNA engagements is largely unaffected by inactive

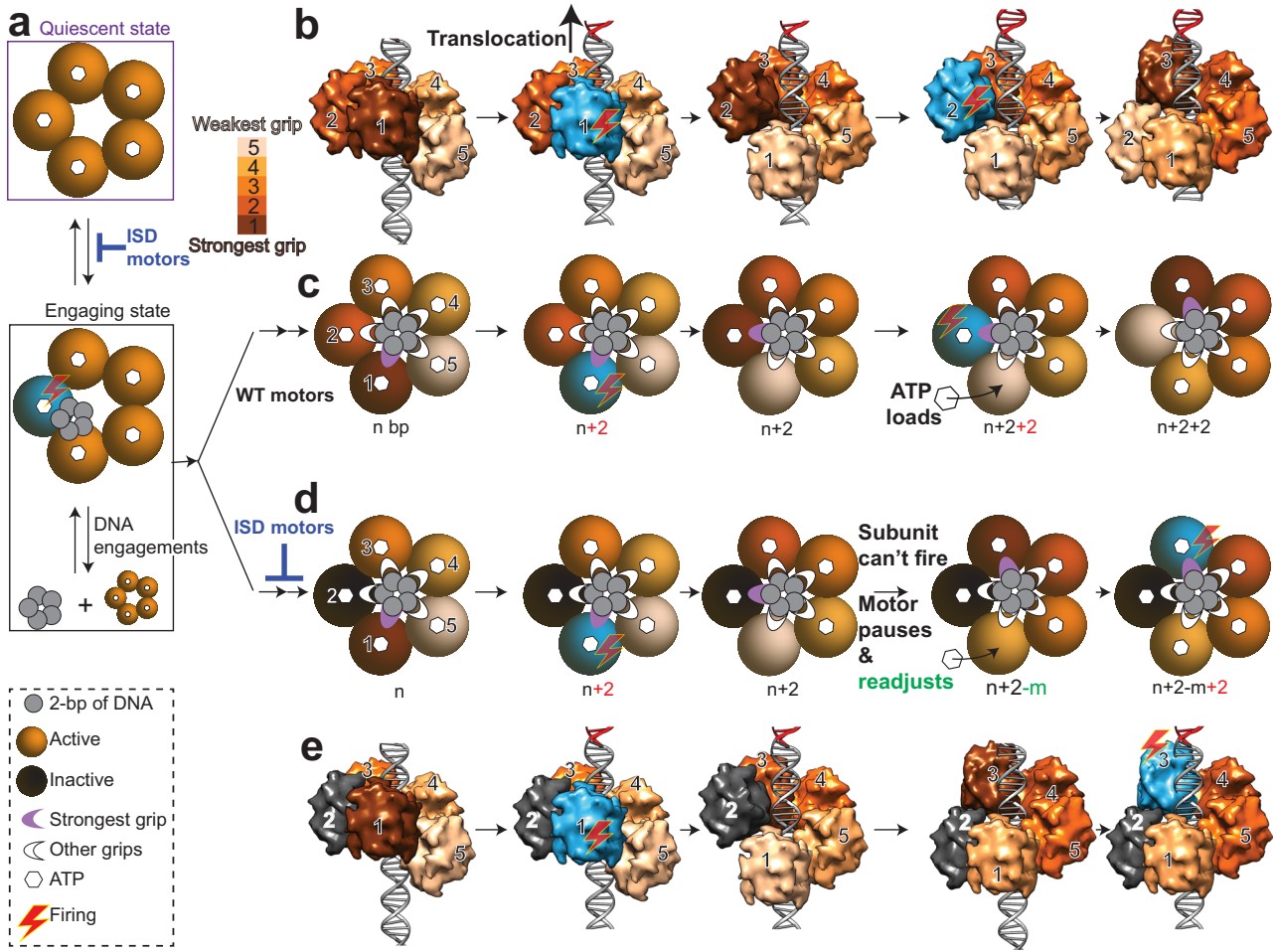

**Fig. 5 A model for DNA translocation and subunit coordination by phage T4 DNA packaging motor. a** The pentameric gp17 DNA packaging motor (five brown circles) alternates between a quiescent state and a DNA engaging active state when exposed to short 45-bp oligonucleotide DNA substrate (gray circles), with the ISD motor spending longer time in quiescent state than the WT motor (shown by blue deleterious symbol). **b** The pentameric motor adapts to DNA helix by interacting at ~2-bp intervals, leading to different subunits binding at different strengths, as shown by increasing color intensity with increasing strength of the grip. The surface view of the putative DNA-binding C-terminal domain of gp17 is used to illustrate DNA–motor interactions. Shown are the side views with the camera facing subunit #1. See Supplementary Video 1 for additional details. The subunit with the strongest grip (#1; cyan) fires ATP, causing 2-bp DNA translocation ($n + 2$). The adjacent subunit (#2) now attains the strongest grip, while the apo subunit (no ATP) re-loads ATP. The #2 subunit now fires, causing another 2-bp translocation ($n + 2 + 2$) and this cycle continues. **c** Cartoon schematic of **b** depicting the motor's top view. DNA is shown in gray in the motor's channel, with each small circle representing 2-bp of DNA. White arches represent DNA-binding grips, while the purple arches represent the strongest grips. Hexagons at the center of each subunit represent bound ATP. **b**, **c** correspond to WT motor, while **d**, **e** correspond to ISD motor. When an inactive motor subunit (black; subunit #2) is encountered (**d**, **e**), the motor pauses and readjusts DNA grip, which might involve backward DNA movement ("*m*" bp). The subunit with the strongest grip (#3) then fires causing DNA translocation and the cycle continues. The shapes depicted in the figure are shown in the bottom left corner of the figure.

subunits (Fig. 3b, right panel). But the overall frequency of DNA engagements by the motor decreases with inactive subunits because the quiescent state is longer lived in ISD motors (Fig. 3c).

Most (~80%) DNA engagements in the active state fail to lead to encapsidation regardless of motor composition (Fig. 3d, left panel). These transient events must correspond to DNA coming into loose contact with the motor and then dissociating. In order for productive packaging to initiate, an additional transition must occur for the DNA to engage tightly with the motor channel, preventing its dissociation against opposing diffusive forces. A decrease in the fraction of long-lived engagements in ISD motors (Fig. 3d, right panel) suggests that this transition is ATP dependent. Furthermore, the requirement for tight DNA engagement poses a significant barrier when occurring from a free DNA end, as it is more likely to slip out of the motor than a stretch of DNA that is already stabilized by multiple subunit contacts. This may account for the high failure rate of initiation. The higher diffusion

coefficients of shorter DNAs may also explain why shorter DNAs tend to be less efficiently packaged than longer DNAs even in bulk assays (Supplementary Fig. 12).

It is also possible that these initial events are facilitated by the "small terminase" gp16[37]. Gp16 and other phage small terminases are essential for packaging in vivo because the newly replicated viral genome in most cases is concatemeric and requires a double-stranded DNA cut in order to generate an end and initiate DNA translocation. The small terminases are reported to interact with DNA[6,7]. This interaction and its association with the large terminase forms a holo-terminase complex that is essential for this process. Therefore, although gp16 is not required for packaging in vitro because the packaging substrate is already cut, it might still be important for the most efficient DNA engagements.

After successful packaging initiation, the presence of inactive subunits does not abolish packaging, suggesting that they can be repeatedly bypassed. Nevertheless, this bypassing causes defects in

packaging, such as slower start, reduced packaging velocity, and more frequent pauses, and unpackaging events than in the WT motor (Fig. 4). How ISD motors bypass inactive subunits rather than arresting translocation, as is known to occur in strictly coordinated motors, remains an interesting question.

Recent structural data provide important constraints on the mechanism of DNA translocation in packaging motors. X-ray and cryo-EM structures show that gp17, and other packaging motor proteins from phages and viruses, consists of two domains, an N-terminal ATPase domain and a C-terminal DNA binding domain connected via a flexible hinge[16,19,20,34]. ATP hydrolysis in gp17 causes a ~7-Å conformational transition between the two domains, proposed to drive a ~2 bp movement of the bound DNA into the capsid[11]. In this model, the motor subunits around the ring take turns and sequentially hydrolyze ATP, causing the domains move in a piston-like fashion, akin to a five-cylinder engine, translocating DNA into the capsid (Video in ref. [11]). Recent cryo-EM data indicate that the packaging subunits of ϕ29 are arranged in a helical configuration[22] tracking the DNA helix, similar to that observed in the structures of ringed hexameric helicases[38,39]. As a result, multiple subunits around the ring are likely to make contacts with the DNA every 2 bp, at the same or similar DNA elements. This is consistent with our recent studies on DNA gripping by single T4 motors, which suggest that multiple subunits, probably all five, when occupied with ATP, bind DNA to generate the grip necessary to hold onto it[35,36]. Thus, the motor–DNA grip consists of one pentameric motor and one helical length of DNA (Supplementary Video).

Based on the above considerations and the single-molecule measurements reported here, we propose a model for DNA translocation (Fig. 5b, c and Supplementary Video). The model requires that the motor subunits must be flexible enough to adjust and generate an asymmetric "nucleosome-like" helical motor–DNA complex; the helical organization is also expected to lead to different motor subunits binding DNA with different strengths. The asymmetric interactions at the symmetry-mismatched portal vertex, a conserved feature in phages and viruses, would also impart flexibility to attain this configuration[40,41]. Consequently, one of the motor subunits (#1) binds DNA with the largest affinity and the rest in decreasing order with subunit #5 with the least affinity due to the expected strain on the binding elements. Subunit #1 with its strongest grip on DNA is triggered to fire ATP hydrolysis, resulting in 2-bp translocation of DNA. Subunit #2 then attains the strongest affinity, while subunit #1 in an ATP-free apo state with the least affinity to DNA re-loads ATP. Subunit #2 then fires, causing another 2 bp of packaging, repeating the next cycle of translocation, ATP reloading, and DNA gripping steps.

Whenever the DNA encounters an inactive subunit (Fig. 5d, e), we propose that the T4 motor shows flexibility to pause and to readjust the DNA registry, through micro-disengagement that may also involve backward DNA movement, until it recovers the right DNA grip and conformation to continue translocation. These micropauses and grip re-adjustments are likely too fast to resolve individually in our assays but are reflected as an overall reduction in the mean packaging velocity. Furthermore, on some occasions, the pauses are longer and lead to detectable unpackaging, a phenomenon characteristically observed with the ISD motors but not with the WT motors. Similar unpackaging events were frequently observed in our previous studies when one or more motor subunits are unoccupied with ATP, which appear to be equivalent to ATPase impaired subunit(s) in the present study[36]. A Monte Carlo simulation based on this model was able to faithfully recapture the motor's packaging velocities with/without inactive subunits (Supplementary Fig. 12). The simulation also showed that lower quanta of average DNA slips minimizes the difference between the motor with varying number of inactive

subunits (Supplementary Fig. 12). We suspect that unpackaging represents events where the motor partially loses its grip on the DNA in the process of readjustment that, under an applied force, leads to loss of DNA registry with the motor and DNA release. When the DNA grip and registry is restored, the motor is reset and packaging continues. Similar unpackaging events were frequently observed in our previous studies[36] when one or more motor subunits are unoccupied with ATP, which appear to be equivalent to ATPase impaired subunit(s) in the present study.

Our study highlights that, while the basic packaging mechanism and the structural and functional features of genome packaging motors are well conserved in phages and viruses[6,7], significant distinctions exist in certain mechanistic details, such as the motor coordination. While the ϕ29 motor uses a highly coordinated, dwell-burst, two-phase mechanism that is controlled at the whole motor level, the T4 motor seems to have adapted a flexible mechanism that is controlled at the individual subunit level. Consequently, while the T4 motor is able to skip one, two, or even three inactive subunits by re-adjusting its DNA grip, the ϕ29 motor could at best tolerate one inactive subunit by making it a special non-translocating subunit and re-setting the dwell-burst cycle accordingly[30]. Any more than one inactive subunit leads to arrest of packaging since both ATP binding and ATPase firing of a motor subunit are strictly dependent upon coordinated activities of the previous subunit. Thus T4 phage has apparently evolved a flexible motor by introducing a degree of independence for each subunit such that it can carry out the central tasks—ATP loading, DNA gripping, and ATP hydrolysis—without strict dependence on its neighbor. While this mechanism can lead to more frequent pauses and slips, as has been observed (Fig. 4)[15,35,42], we speculate that it would eliminate the dwell phase and make T4 motor a faster motor that is able to skip subunits as well as pause and readjust when obstacles are encountered, such as a defective subunit or when packaging the newly replicated T4 DNA, a highly concatemeric and branched structure. Importantly, this mechanism would confer survival advantages to T4 phage such as packaging its ~171-kb genome, ~8.5 times longer than that of the ϕ29, in about the same amount of time as the ϕ29 phage[43].

## Methods

**Preparation of T4 heads**. Empty phage T4 heads (capsids) for packaging experiments were prepared according to the protocol described previously[44]. Briefly, *E. coli* P301 (sup-) cells (500 ml) were infected with *10am13am* mutant T4 phage and lysed in 40 ml of Pi-Mg buffer (26 mM $Na_2HPO_4$/68 mM NaCl/22 mM $KH_2PO_4$/1 mM $MgSO_4$, pH 7.5) containing 10 μg/ml DNase I and chloroform (1 ml) and incubated at 37 °C for 30 min. The emptied heads thus produced were purified by differential centrifugations involving low-speed (6000 × g for 10 min) followed by high-speed (34,000 × g for 45 min) centrifugations. The final heads pellet was resuspended in 200 μl of Tris·Mg buffer (10 mM Tris·HCl, pH 7.5/ 50 mM NaCl/5 mM $MgCl_2$) and purified by CsCl density gradient centrifugation. The major band containing empty capsids, appearing at about 1/3 from the bottom of a 5-ml gradient, was extracted and dialyzed overnight against Tris·Mg buffer. The capsids were further purified by DEAE-Sepharose ion-exchange chromatography. The peak fractions containing pure capsids were concentrated and stored at −80 °C.

**Cloning and expression of gp17 mutants**. The T7 promoter containing *E. coli* expression plasmid vector pET-28b (Novagen, MA) was used to construct all the gp17 WT and mutant clones[45]. DNA fragments containing the gene of gp17 or ATPase defective mutant Q163A-gp17 were amplified by the polymerase chain reaction (PCR) using specific primers containing appropriate restriction site(s). Appropriate mutations were introduced using mutant primers by the overlap PCR strategy described previously[33]. The sequences of the primers, along with that of 45-bp DNA used in single-molecule imaging, is available in Supplementary Table 1. The PCR products were purified, digested with appropriate restriction enzymes (Thermo Scientific, Waltham, MA), and ligated with pET-28b vector DNA digested with the same restriction enzymes. The ligated DNAs were transformed into *E. coli* XL10-Gold Ultracompetent Cells (Agilent Technologies, Santa Clara, CA) and miniprep plasmid DNAs were prepared from single transformants. The accuracy of the cloned DNA was determined by DNA sequencing of the entire

insert and the flanking sequence (Retrogen, San Diego, CA). The resulting plasmids had gp17, Q163A-gp17 coding sequences fused in-frame with the 23 aa vector sequence containing a hexa-histidine tag at the N-terminus.

To construct SNAP-tagged gp17, the SNAP-tag fragment was digested from the pSNAP-tag (T7)-2 vector (New England Biolabs), inserted into pET-28b plasmid resulting in a construct referred as pET-SNAP. The full-length WT-gp17 was amplified from pET15-gp17 constructed previously and inserted into pET-SNAP to generate SNAP-gp17 recombinant. The insert was translated as a fusion protein in which SNAP-tag was fused to a hexa-histidine tag followed by full-length WT-gp17. Using this construct as a template, the ATPase defective mutant SNAP-Q163A-gp17 was constructed using the overlap-PCR strategy and appropriate mutant primers. The presence of DNA inserts and their orientation with respect to the promoter were tested using restriction enzyme digestion and/or amplification with insert-specific primers. The accuracy of the mutant DNA was confirmed by sequencing the entire insert DNA (Retrogen, Inc., San Diego, CA). The mutant plasmids were then transformed into *E. coli* BL21 (DE3) pLysS-competent cells (Stratagene) for protein expression and purification. All the recombinant DNAs were then transformed into the expression strain *E. coli* codon-plus BL21 (DE3) RIPL (Stratagene) for IPTG-induced overexpression of the recombinant proteins.

**Protein expression and purification**. The recombinant proteins were purified according to the basic protocol described as follows[45]. *E. coli* strain BL21 (DE3) pLysS or *E. coli* codon-plus BL21 (DE3) RIPL cells containing His-tagged WT-gp17 and Q163A-gp17 mutant clones were induced with 1 mM IPTG at 26–30 °C for 2.0–3.5 h. The solubility of the overexpressed proteins was tested using the B-PER reagent according to manufacturer's instructions (bacterial protein extraction reagent; Pierce). The IPTG-induced *E. coli* cells were harvested by centrifugation (4000 × g for 15 min at 4 °C) and resuspended in 50 ml of HisTrap binding buffer (50 mM Tris·HCl, pH 8.0/20 mM imidazole/300 mM NaCl). The cells were lysed using an Aminco French press (Thermo Fisher Scientific Inc., Waltham, MA). The cell lysates were centrifuged at 34,000 × g for 20 min at 4 °C and the supernatant containing the His-tagged fusion protein was loaded onto a HisTrap column (AKTA-prime, GE Healthcare). The protein was eluted with a buffer containing 20–500 mM linear imidazole gradient. The peak fractions containing pure gp17 were concentrated and further purified using size-exclusion chromatography using Hi-Load 16/60 Superdex-200 (prep-grade) gel filtration column (GE Healthcare) in a buffer containing 20 mM Tris·HCl (pH 8.0) and 100 mM NaCl. The SNAP-gp17 and SNAP-Q163A-gp17 were overexpressed and purified using the same protocol as above except that SNAP-gp17 purification required the addition of reducing reagent such as 1 mM dithiothreitol (DTT) or 0.5 mM TCEP to prevent nonspecific disulfide bond formation. The peak fractions were concentrated by Amicon ultra-15 centrifugal filtration (MW 10 K cut off; Millipore, Temecula, CA) and stored at −80 °C.

**SNAP labeling of T4 motors**. Frozen aliquots of SNAP-gp17 and ATPase-inactive SNAP-Q163A-gp17 proteins in gel filtration buffer (50 mM Tris·HCl pH7.5, 100 mM NaCl, 1 mM DTT) were put on ice and mixed with SNAP-tag fluorophores (SNAP-Surface 549 purchased from New England Biolabs) dissolved in dimethyl sulfoxide and 0.1% Tween-20. The labeling mixture was incubated with gentle rotation at room temperature for 2 h in the dark. The dye concentration was 2–3-folds higher than protein concentration. Excess free dye was removed by size-exclusion chromatography using the Superdex® 200 Increase 10/300 GL (prep-grade) gel filtration column (GE Healthcare) and gel filtration buffer at a flow rate of ~0.4 ml/min for separation of free dye from labeled protein. Fractions of labeled protein were collected, aliquoted, and frozen at −80 °C, protected from light. The labeling efficiency of dye was analyzed by sodium dodecyl sulfate–polyacrylamide gel electrophoresis followed by fluorescence imaging using Gel ChemiDoc system (Bio-Rad, Hercules, CA). The concentration of SNAP-gp17 was quantified using Image Quant software normalized to standard protein markers, and dye concentration was determined from the absorbance at 550 nm. The labeling efficiency of different batches ranged from 70 to 90%, which are likely to be slight overestimates as the *complete* removal of free dyes may not have been achieved. The labeling efficiency of the Cy3 on the SNAP-gp17, used for stoichiometry calculations (shown in Fig. 1), was 88%. The labeling efficiency, as determined from the fitting for stoichiometry of Fig. 1, was ~66%. This discrepancy between 88 and 66% may be due to the overestimation of the labeling efficiency described above.

**In vitro DNA packaging**. In vitro DNA packaging assays were performed by a previous described procedure[46]. The purified gp17 protein (~1.5 μM) was incubated with empty capsids purified as above (~2 × 10^10 particles) and DNA [300 ng of 50- to 766-bp ladder DNA (New England Biolabs, Ipswich, MA) or linearized 2.8- or 6-kb plasmid DNA], and packaging buffer containing 30 mM Tris·HCl (pH 7.5), 100 mM NaCl, 3 mM MgCl₂, and 1 mM ATP for 5 min of short time incubation, or for 30 min of long-time incubation at 37 °C. Unpackaged DNA was degraded by adding DNase I (Sigma-Aldrich) by incubating this mixture for 30 min at 37 °C, and the encapsided DNase-I-resistant DNA was released by treatment with proteinase K. The retrieved encapsidated DNA was then analyzed by polyacrylamide (4–20% gradient) or agarose (0.8%) gel electrophoresis. Each experiment included one to several negative controls that lacked one of the

essential packaging components: gp17 or ATP. A standard lane containing a small amount of the DNA substrate used in the packaging reaction allowed for quantification of packaged DNA using the Gel ChemiDoc MP imaging system (Bio-Rad, Hercules, CA). In the DNA packaging experiments involving doping with inactive subunits, the purified WT-gp17 was mixed with purified inactive Q163A-gp17, with increasing fractions of the inactive subunit. The packaging efficiencies were quantified and normalized to the DNA density of the sample in which only WT-gp17 was added for packaging.

**Mathematical model of the ensemble packaging activity of the hetero-pentameric T4 motor**. A binomial distribution of the form shown below was used to describe the distribution of hetero-pentameric motors containing inactive subunits:

$$f(k, p) = \frac{n!}{k!(n-k)!}(1-p)^k p^{n-k} \tag{1}$$

where $f(k,p)$ is the probability that a pentameric motor ($n = 5$) contains $k$ active subunits, given the probability $p$ that an inactive subunit is present in the motor. Inactive subunits and WT subunits are presumed to have the same probability of being present in the motor. $p$ was calculated as the molar ratio of inactive subunits to all subunits present in the assay:

$$p = \frac{\text{Inactive}}{\text{Inactive} + \text{WT}} \tag{2}$$

To model the activity of hetero-pentameric motors, we assumed that there is a minimum number of active subunits, $k_{min}$, required for packaging activity and that each active subunit contributes a fraction $1/n$ of the total activity. The predicted packaging activity, normalized by the maximum possible activity when all subunits are WT, is given by:

$$\text{Activity}(k_{min}, p) = \frac{1}{n}\sum_{k=k_{min}}^{n} kf(k, p) \tag{3}$$

If the subunits are strictly coordinated, then one inactive subunit is sufficient to abolish packaging activity entirely, and $k_{min} = 5$. If the subunits are completely independent, then a single active subunit is sufficient to drive packaging, and $k_{min} = 1$. Intermediate coordination mechanisms require 2–4 active subunits. The ensemble packaging activity for different $p$ values was compared to these different models of coordination (Fig. 2e).

**Preparation of channels and constitution of T4 motor for single-molecule fluorescence imaging**. The T4 motor complex containing the empty head and motor was assembled on a PEG passivated surface for imaging its activity (Supplementary Fig. 2a). A buffer condition optimal for packaging activity i.e., 50 mM Tris·Cl (pH 7.5), 5 mM MgCl₂, 1 mM spermidine, 1 mM putrescine, 60 mM NaCl, and 5% (wt/vol) PEG, 0.2 mg/ml bovine serum albumin (BSA), 3 mM Trolox, and an oxygen scavenging system (0.8% dextrose, 0.1 mg/ml glucose oxidase, and 0.02 mg/ml catalase) was used as the imaging solution for all the experiments, also referred as "Imaging Buffer-BSA." To achieve specific surface immobilization of T4 motor for visualization at the scale of single motor, clean quartz slides and glass coverslips were surface-passivated with PEG (Layson Bio (Arab, AL)) and 1% (wt/wt) biotinylated PEG (Layson Bio (Arab, AL)). After preparing the channels using the quartz slides and glass coverslips, NeutrAvidin (Thermo Scientific) was added to the channel surface (0.2 mg/ml), followed by incubation with biotinylated protein-G (25 nM; Rockland Immunochemicals, Pottstown, PA) for 30 min. Following surface immobilization of protein-G, mouse polyclonal anti-T4 antibody (15 nM) was added to the channel surface and incubated for 30 min, which allowed for the binding of the anti-T4 antibody. Anti-T4 antibodies were raised in mice in compliance with all relevant ethical regulations for animal testing and research recommended by NIH (the Guide for the Care and Use of Laboratory Animals) and approved by the Institutional Animal Care and Use Committee of the Catholic University of America (Washington, DC) (Office of Laboratory Animal Welfare assurance number A4431-01)]. The channel was then washed with Imaging Buffer-BSA to remove the unbound antibodies.

Meanwhile, a separate reaction buffer to pre-assemble the gp17 packaging motor protein onto the capsid was prepared by mixing the heads (1 × 10^10 particles per slide channel), ATP-ɣS (1 mM), 120 bp DNA (200 nM), and gp17 monomer (2–5μM) in packaging buffer with PEG (50 mM Tris·HCl, pH 7.5, 100 mM NaCl, 5 mM MgCl₂, 5% w/v PEG-2000) followed by 15 min incubation in the dark at the room temperature (0.5–5 μl motors per slide chamber). The poorly hydrolyzed ATP-ɣS analog and 120 bp unlabeled DNA were used to stabilize and enhance the efficiency of assembly of functional T4 motors[15]. The 120-bp DNA is bound to the motor in the presence of ATP-ɣS, but not packaged. It is likely that some of these molecules are packaged when ATP is added and before the first Cy5-labeled DNA is engaged with the motor. These pre-assembled head–motor complexes were then added to the channel and incubated in the antibody-coated channel surface for 30 min, and the chamber was then washed with Imaging Buffer-BSA to remove unbound capsids. Following all the steps described above, the T4 packaging activity was then imaged using TIRF microscopy through objective and prism-type illumination. All the imaging was done at room temperature. Cy5-labeled 45-bp DNA oligonucleotides were purchased from Integrated DNA Technologies (Coralville, IA).

**Single-molecule fluorescence detection and data analysis**. The fluorescence emission from the Cy3-labeled, immobilized motors carrying out Cy5-DNA packaging was refocused onto an EMCCD Andor Camera. The final emission was split into two separate channels for imaging Cy3 (for Cy3-labeled subunits) and Cy5 (for Cy5-labeled DNA). Cy3–Cy5 imaging was carried out in one of two ways. (1) Motors were incubated a few minutes in the dark with the addition of 1 mM ATP and 2.5 nM Cy5-DNA into the chamber to allow packaging to proceed for a sufficient time. DNase I was then added to digest and flush the unpackaged DNA, followed by Cy3–Cy5 imaging. (2) Alternately, imaging was done in real time upon the addition of 1 mM ATP and 2.5 nM Cy5-DNA. Cy3 imaging for Cy3 stoichiometry was done with Cy5 imaging either before or after it. The order of Cy3/Cy5 imaging did not affect the experiments. The time trajectories of Cy3 and Cy5 fluorescence intensities were extracted from the movies and then further analyzed using customized MATLAB programs. For some analysis, the trajectories were smoothed using the nonlinear forward–backward filter, with filter parameters of $n = 4$, $M = 3$, and $P = 20$, to aid the analysis[47]. Some representative trajectories shown in the figures have also been smoothed using this filter. The time resolution for all the experiments was 100 ms unless stated otherwise.

**Stoichiometry calculations and binomial fitting**. The photobleaching steps of the Cy3 signal were pooled together from multiple fluorescence time trajectories. The distribution of the photobleaching step counts is representative of the stoichiometry of the active/inactive subunits of the T4 motor as a particular subunit is labeled with Cy3. To obtain the stoichiometry of active subunits in the motor constituted with 100% Cy3-labeled active subunits, the raw distribution of the Cy3 photobleaching step counts was compared to a binomial distribution of the form shown below:

$$s(k) = \frac{n!}{k!(n-k)!} p^k (1-p)^{n-k} \qquad (4)$$

where $s(k)$ is the probability of detecting $k$ Cy3-labeled subunits from photobleaching steps in a motor of stoichiometry $n$, given the Cy3 labeling efficiency $p$. The value of $n$ that gives the best fit of the raw distribution was taken as the stoichiometry of the motor.

**Number of Cy5 spots and the number of packaged DNA in the Cy5 spots for motor with quantifiable number of active/inactive subunits**. The efficiency of DNA packaging by a particular combination of subunits constituting the T4 motor was quantified by determining the average number of fluorescent spots per area from multiple imaging areas for each sample. To probe the efficiency further, since a single fluorescent spot could represent multiple Cy5-DNA molecules packaged inside the capsid, a more detailed approach was used. In this case, the count of Cy5 photobleaching steps acquired from the time trajectories of long duration movies was used to quantify the number of packaged Cy5-DNA inside the capsid. For each spot, representing a single motor, its stoichiometry of the active/inactive subunits was determined by the number of Cy3 photobleaching steps. The list of the counts of packaged Cy5-DNA, for motors with different number of active/inactive subunits, is pooled together from multiple time trajectories.

**Average time to engage DNA and average duration of DNA engagements**. The fluorescence time trajectories showing Cy5 intensity spikes, showing real-time DNA engagements by the motor, were manually selected using custom MATLAB scripts. These trajectories were then idealized using a previously reported statistical model developed for idealization and analysis of single-molecule trajectories[48]. After idealization, the Cy5 spikes were automatically classified as transient, semi-transient, or long-lived DNA engagement events. Each long-lived DNA engagement event was taken as the successful packaging of a DNA. The dwell times between the DNA engagements were selected from multiple trajectories. The survival probability distributions of these dwell times were fit with appropriate exponential decay functions to estimate the average time it takes the motor to engage DNA.

**Preparation of DNA substrate for optical trapping assay**. The DNA substrate used in the optical trapping assay was a 3400-bp PCR-amplified section of plasmid PBR322 (New England Biolabs). The forward primer (5'-/btn/-ACA GCA TCG CCA GTC ACT AT-3') was modified with a 5' biotin for attachment to streptavidin beads; the reverse primer (5'-CAA CAA CGT TGC GCA AAC T-3') was unmodified (primers from Integrated DNA Technologies). The PCR product was purified by a Qiagen PCR Purification Kit.

**Preparation of stalled packaging motor beads for optical trapping assay**. Streptavidin beads were used to bind biotinylated DNA on which stalled T4 motors were assembled. In all, 50 μl of 1% w/v 800-nm streptavidin-coated polystyrene beads (Spherotech Inc.) were washed three times in 200 μl 1× phosphate-buffered saline (PBS) buffer and centrifuged at 6723 × $g$, discarding the supernatant each time. After the final wash step, the beads were resuspended in 50 μl 1× PBS buffer. In all, 5 μl of the washed streptavidin beads and ~110 ng of 3400-bp biotinylated dsDNA substrates were mixed in 1× binding buffer (10 mM Tris·HCl, 1 M NaCl) to

make up a total volume of 20 μl. The mixture was incubated at room temperature for 3 h. The mixture was then washed with 50 μl 1× packaging buffer to remove any unbound DNA and gently resuspended with 30 μl water and 5 μl of 10× packaging buffer. Next, stalled packaging motors were assembled by incubating 2 μl of 20 μM WT gp17 monomers (or 1 μl of 20 μM WT gp17 + 1 μl of 20 μM Q163A gp17 for inactive subunit doping experiments), 2 × 10^10 T4 heads, 7 μl of DNA-coated beads, and 1.5 μl 10 mM ATP-γS. Note that no 120-bp DNA was used in the assembly of these complexes. The mixture was incubated on a rotator at room temperature for 1 h. It was then diluted into 500 μl of 1× packaging buffer containing 1 mM ATP-γS and transferred to another empty Hamilton syringe.

**Preparation of antibody-coated beads for optical trapping assay**. Anti-T4 antibody-coated beads were used to bind T4 heads. In all, 1% w/v 800 nm Protein G-coated polystyrene beads (Spherotech Inc.) were washed in the same way as the streptavidin beads. Ten microliters of the purified beads were mixed with 2 μl of undiluted serum containing polyclonal anti-T4 antibodies and incubated at room temperature for 45 min, followed by 3 washes with 50 μl 1× PBS buffer. The antibody-coated beads were finally resuspended in 10 μl 1× packaging buffer and stored at 4 °C for up to 1 week. For each experiment, 2 μl of antibody-coated beads were diluted into 500 μl 1× packaging buffer and transferred into an empty Hamilton syringe.

**Dual optical traps and laminar flow chamber**. Single-molecule optical trap packaging was conducted on high-resolution dual-trap optical tweezers, as previous described[49,50]. The instrument and data collection were controlled by custom Labview programs and Matlab code.

In order to control the assembly of the packaging motors and initiation of packaging, we used a laminar flow chamber with multiple channels[51]. The chamber was constructed by melting a patterned piece of parafilm between two glass coverslips, one of which had holes drilled for entry and exit of samples by syringe pumps. Sample chambers were designed with 4 channels (Fig. 4a): 2 inner channels that merged together but maintained a sharp interface (<0.5 mm)[52] due to the laminar flow, and 2 outer channels separated by parafilm but each connected to the middle channels by a glass capillary to allow for exchange from one channel to the other. Streptavidin beads coated with arrested motors and anti-T4 antibody-coated beads were pumped through the two outer channels, respectively, and leaked into the inner channels. The inner channels contained 466 μl 1× packaging buffer (50 mM Tris·HCl, pH 7.5, 100 mM NaCl, 5 mM MgCl₂), 5 μl Pyranose Oxidase oxygen scavenging system, 25 μl 20% glucose, and 4 μl of substrate: either ATP at a final concentration of 2 mM in the top inner channel or ATP-γS at a final concentration of 0.4 mM in the bottom inner channel (Fig. 4a). The oxygen scavenging system, used to reduce photodamage to the packaging motors and to ensure the DNA tethers had long lifetimes[53], was prepared by dissolving 2.9 mg of Catalase (*Aspergillus niger*) and 5.8 mg of Pyranose Oxidase (*Coriolus* sp.) in 200 μl of T50 buffer (10 mM Tris·HCl pH 7.5, 50 mM NaCl). All samples and buffer stored in Hamilton syringes were flowed into the chamber via polyethylene tubing and controlled by syringe pumps.

**Optical trap assay procedure**. In a typical experiment, an anti-T4 antibody-coated bead was first trapped near the top capillary in the ATP channel (Fig. 4a, step ①). Then the traps were moved to the lower capillary in the ATP-γS channel, and a bead coated with arrested packaging motors on DNA was captured in the other trap (Fig. 4a, step ②). A tether was formed when beads were moved close to each other for 1 s and separated (fishing). Successful tethering was detected by a rise in force (>5 pN) as the beads were moved apart (Fig. 4a, step ③). After a correct tether formed, a constant force of 5 pN was applied in order to keep DNA extended between the beads. Then the trapped bead pair was moved from the ATP-γS channel to the ATP channel. Packaging could start at a constant load force and was measured from the decrease in DNA extension between the two beads, as DNA was translocated into the capsid by the packaging motor (Fig. 4a, step ④).

Once a tether formed, its elastic response (force–extension curve) was measured prior to initiating packaging. Traces were selected or discarded based on inspection of the trap calibration parameters and the elastic behavior of each tether. Specifically, each force-extension curve was compared to the extensible worm-like chain model[54] with a persistence length of 50 nm, a stretch modulus of 1000 pN, a distance per base pair of 0.34 nm, and contour length of 1156 bp. Due to the uncertainty of the size of beads and the orientation of the heads bound to beads, we found there could be an extension offset between this force–extension curve and the model. Typically, we found that most tethers had force–extension curves within −100 nm to 50 nm of the model, and tethers outside this range were discarded. In addition, we calculated the residuals between the extension-offset-corrected force–extension curve and the model above. Tethers for which the extremum and variance in the residuals exceeded a global threshold were discarded from analysis.

**Packaging dynamics analysis**. Custom Matlab code was written to distinguish pauses from packaging and to determine dynamic parameters for each selected packaging trace. The velocity was calculated from the slope of a linear fit to the data over a running 0.1-s window. A velocity threshold was assigned according to the variance in velocity, below which the time points were marked as potential pauses.

Time intervals of duration >0.1 s in which all time points were below the velocity threshold were recorded as a pause. The pause frequency was determined by dividing the number of pauses by the sum of packaging lengths during each packaging interval. Then the pauses were removed from the trace and the remaining packaging intervals were catenated. The mean packaging velocity was determined as the average of the velocity of the catenated trace.

**Statistics and reproducibility**. For all the experiments reported with error bars, 2–5 replicate and reproducible experiments were performed to get statistics of mean and standard deviation. Values represent mean. Error bars throughout represent standard deviation.

*Modeling and simulation*. The packaging and slipping of the DNA into the capsid by the T4 motor was modeled as a one-dimensional (1-D) walk. The kinetic Monte Carlo simulations of such a 1-D walk was carried out using the Gillespie Algorithm. At any step of the simulation, if the subunit with the strongest grip (responsible for controlled 2-bp packaging) is not an inactive unit, the motor had a probability of packaging by 2 bp ($p_{package}$) indicating the controlled and ATP hydrolysis-driven event. Alternatively, the motor had a probability of pausing and slipping (unpackaging) by $n$ bp ($p_{pause\&slip}$). Since slipping is an uncontrolled event, the extent of each slip ($n$ bp) was Poisson-distributed, and the value of $n$ was randomly chosen from the exponential distribution of the form: exp ($\mu$), where $\mu$ is the average length of the slip, taken to be 0.5. At each step in the simulation, there were only two possibilities: DNA moving forward in the form of packaging (given by $p_{package}$) or backward in the form of slipping (given by $p_{pause\&slip}$). $p_{package} = k_{package}/(k_{pause\&slip} + k_{package})$ and $p_{pause\&slip} = 1 - p_{package}$, where $k_{package}$ and $k_{pause\&slip}$ are the rates of packaging and pausing & slipping, respectively, as described in Fig. 5. The value of $p_{pause\&slip}$ depended on the number of active subunits/inactive subunits in the motor. For the simulations, $p_{pause\&slip}$ was 0.65, 0.75, and 0.75 for the motor with 0, 1, and 2 inactive subunits to match the experimental data of mean packaging velocity (Fig. 4) and fraction of successful encapsidation events (Fig. 3). If the subunit with the strongest grip is inactive, then $p_{package}$ was taken to be 0.

At each step of the simulation, a random number was generated between 0 and 1. For the motor with 0 or 1 or 2 inactive subunits, If the random number was $>p_{pause\&slip}$, i.e., indicating packaging, then the length of the unpackaged DNA length was reduced by 2 bp. If the random number was $<p_{pause\&slip}$, i.e., indicating pause and slipping, then the length of the unpackaged DNA length was increased by $n$ bp where the value of $n$ was randomly chosen from the exponential distribution as described above.

At each step of the simulation, the time expended in the packaging or pause&slip was also taken to be exponentially distributed and taken from the exponential distribution of the form: exp ($t_{package}$) or exp ($t_{pause\&slip}$), where $t_{package}$ is the average time to package 2 bp and $t_{pause\&slip}$ is the average time of pausing and slipping. For the simulations, values of $t_{package}$ and $t_{pause\&slip}$ were 10 arbitrary units (arb. units) and 2 arb. units, respectively, with the tacit assumption that a controlled event of packaging could take longer times than the fast/uncontrolled pause and slipping event. The values $t_{package}$ and $t_{pause\&slip}$ did not significantly impact the simulated trajectories, in contrast to $p_{pause\&slip}$, which was far more critical.

Trajectories (unpackaged length vs. time in arb. units) whose unpackaged DNA length reached 0 indicated successful packaging. The trajectories whose unpackaged DNA length exceeded the length of the DNA indicated failed packaging. Such trajectories imply that the entire DNA would have slipped enough to dissociate from the motor and capsid. Simulations describing the packaging of ~3400 bp long DNA were started assuming that 30–40 bp of the ~3400 bp was already encapsidated, to prevent DNA from completely slipping out of the capsid. The pre-packaging of 30–40 bp DNA inside the capsid ensured that the ~3400 bp long DNA could never wholly slip out of the capsid, similar to the experimental observation. The mean velocity $v$ of packaging of the motor with 0 inactive subunit was ~2 times that of the motor with 1 or 2 inactive subunits, similar to the experimental data in Fig. 4. The nature of pauses in the motor trajectory with 1 or 2 inactive subunits is also qualitatively similar to the nature of pauses observed in the experimental data of Fig. 4.

**Reporting summary**. Further information on research design is available in the Nature Research Reporting Summary linked to this article.

## Data availability
The data supporting the findings of this study are available from the corresponding authors upon reasonable request. Source data are provided with this paper.

## Code availability
The codes for processing single-molecule fluorescence movies are available here: https://github.com/Ha-SingleMoleculeLab. The codes for processing of Optical trap data are available here: https://gitlab.com/chemla-lab-public-code/2021_NatComm_PhageT4. Any other codes in this study are available from the corresponding authors upon reasonable request.

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

## Acknowledgements

This work was supported by the National Science Foundation grant MCB-0923873 and National Institutes of Health (NIH) grant AI081726 to V.B.R., NIH grant R35 GM122569 to T.H., NIH grant R01 GM118817 to Y.R.C., and National Science Foundation grant PHY 1430124 to T.H. and Y.R.C. We thank Dr. Kiran Kondabagil for providing gp17 mutant clones, Dr. Neeti Ananthaswamy for doing a packaging assay of gp17 proteins, and Dr. Victor Padilla-Sanchez for assistance with Fig. 5 and Supplementary Video.

## Author contributions

V.B.R., T.H. and Y.R.C. directed research. L.D., D.S., V.I.K., S.L., R.V., Y.R.C., T.H. and V.B.R designed experiments. L.D. and D.S. performed TIRF experiments. S.L. and V.I.K. performed optical trap experiments. M.M. and V.I.K. produced capsids and performed bulk packaging experiments. L.D., D.S., V.I.K., S.L., R.V., Y.R.C. and V.B.R. analyzed and interpreted data. D.S., L.D., Y.R.C. and V.B.R. wrote the paper.

## Competing interests

The authors declare no competing interests.
