## [Peer Review File · Nature Communications]

REVIEWER COMMENTS

Reviewer #1 (Remarks to the Author):

In the manuscript by Dai et al., the authors present a series of single-molecule studies to examine the effect of mutant poisoning on the phage T4 DNA packaging motor. Most of the data is from well executed (and presumably very challenging) single particle fluorescence experiments and optical tweezers studies. The main conclusion is that the inclusion of one or more P-loop-mutant ATPases into the motor pentamer disrupts, but does not preclude, packaging initiation and DNA translocation. The authors conclude with a somewhat confusing interpretation that these results indicate that the motor is highly coordinated, but not so coordinated that it cannot overcome the presence of a defective subunit. Although well written, some of the text includes statements that seem to make inferences beyond the strength of the data and end with a mechanism that is self-contradicting. This manuscript will require significant revision to resolve these issues.

Most of my criticisms are not what data is presented or analyzed, but how the results are interpreted and discussed relative to the existing literature.

Detailed comments.

Abstract:

Here and throughout the authors describe the ability of the motor to tolerate defective subunits as “unexpected”. This is quite likely not the case as this is one of the few possible outcomes of such an experiment. This term should not be used.

Introduction:

Line 90. Was the stoichiometry of the T4 motor in doubt? If not, the stoichiometry reported here is a reiteration, but not new.

Line 96. The statement “meaning that a single inactive subunit was sufficient to abolish packaging”, which is leaned on heavily throughout the manuscript, is incorrect. Whereas the bulk ensemble studies with Walker A mutant in lambda suggests a lack of tolerance for a defective subunit, in the paper by Tafoya et al. from the Bustamante group, motors with arginine-finger mutants were shown to package DNA with altered kinetics and stepping. Whereas the Tafoya paper does not quantify the number of mutant subunits like in this manuscript, it is with certainty that the motor can tolerate, in vitro, some defective subunits.

This becomes an issue when the authors discuss their own results in attempting to draw a distinction between the mechanism and degree of coordination reported for other systems and that reported

here for T4. Rather than diverging, to me these mechanisms appear to reinforcing each other, thus the central claim of T4 being somehow different seems inaccurate. The central claim that the data shows a dramatic difference between phage systems is not well supported.

Results:

Line 159. The Q163A mutant is described as impaired in hydrolysis. The authors should clarify whether or not, as a P-loop mutant, it is also defective in ATP binding since failure to bind ATP vs failure to hydrolyze have different implications when trying to interpret this phenotype in terms of what step is blocked by the mutant.

Lines 171-173 and 183. The assertion that any measurable activity in mutant doped rings can differentiate between a “strictly coordinated” and “not strictly coordinated” motor is contrary to how this has been interpreted in other phage packaging systems. The remainder of the paper is spent describing how mutant-doped motors differ in their behavior from wild type, indicative of the fact that all subunits participate in packaging and are thus coordinated in some way. Were the authors to discuss how their results show that coordination in T4 is the same as or different from other motors, and perhaps now better understood, the manuscript would add more to the ongoing discussion in this field.

Line 188. I believe this is meant to say “Q163A”

Lines 260-262. I’m not entirely sure how the wild type complexes are identified compared to the Cy3 spots. Are all Cy5 spots without Cy3 considered wild type? If so, for consistency are only Cy3 spots with DNA analyzed as part of the mutant-containing pool?

Lines 280-281. If ATP-gamma-S allows the motor to bind but not package DNA, would one expect the same from the P-loop mutant? Again, the precise phenotype of the Q163A mutant would clarify this.

Discussion:

Lines 359-364. The argument that T4 is “fundamentally distinct” is not well supported. The authors invoke the idea that T4 is different because it is not “completely uncoordinated”. This is hair splitting, leaves the impression that the phage systems do not inform one another, and misses the opportunity to advance understanding of the packaging mechanism by reflecting on similarities rather than insisting they are different.

Lines 392-end. After having spent the entirety of the manuscript arguing that T4 is distinct from other studied packaging systems, the authors incorporate new structural studies on phi29 into their mechanism. What justifies this comparison? (although, as stated above, their results could be used to support an argument that these systems are indeed similar). The mechanism they proposed

appears processive, ordinal, and therefore coordinated. Again, they argue that their mutant studies show that it doesn't have to be, but confusingly they have presented a mechanism that is coordinated. This is very confusing.

Line 440. The argument that the inability to abolish activity by modifying the motor "would confer survival advantages" suggests that the motor "evolved" to tolerate defects. Whereas I can understand that the motor could be seen to require flexibility to operate under a range of conditions, just because it can tolerate a mutant subunit doesn't mean it evolved to be able to do so. If this were the case, where are the phages with defective ATPases?

Reviewer #2 (Remarks to the Author):

In this work, Dai et al. investigated the intersubunit coordination of phage T4 DNA packaging motor using single-molecule TIRF and optical tweezers. By counting the number of inactive subunits in the ring motor via photobleaching and correlating it to the packaging behavior, the authors concluded that the motor consists of five subunits and can tolerate one inactive subunit, even though ISD impairs DNA engagement and translocation, arguing against a strictly coordination model. Overall, the data are of high quality and for the most part rigorously analyzed. The combination of multiple single-molecule techniques is a strength of the paper. However, I have the following concerns that should be addressed before the paper is acceptable for publication.

(1) In both Abstract and Introduction, the authors mentioned that they "precisely count the number of active/inactive motor subunits". This is misleading. Because of <100% labeling efficiency, the number of inactive subunits was inferred by statistical probabilities, which was done for Fig 1c. However, this was not done for Fig 2g (one photobleaching step doesn't necessarily mean one inactive subunit, etc.). Can the authors take the labeling efficiency into consideration and reanalyze the data? If not, at the minimum this caveat should be clearly stated.

(2) In Fig 2e, why only 1- and 2-ISD data (red) are plotted? 0- and 3-ISD data are also measured from Fig 2f and 2g. Including all would make comparison with the bulk measurement (blue) and theoretical curves more conclusive.

(3) How did the authors distinguish between stable, semi-transient, and transient engagements? The distributions of their lifetimes inevitably overlap, especially for transient and semi-transient populations. A more detailed analysis should be conducted to show that the semi-transient population indeed exists.

(4) The optical tweezers experiments showed heterogeneous velocities, pausing frequencies, and waiting times before start. This is likely due to the different number of inactive subunits (one vs. two, for example) in the ring, which was not monitored in these experiments. The authors should comment on this issue and discuss why the heterogeneity in ISD motors is or is not higher than in WT motors.

(5) The amount of unpackaging seen in Fig. 4b is up to hundreds of base pairs, which seems at odds with the model depicted in Fig 5, where readjustment of the motor's DNA grip takes no more than a few bps.

(6) The authors stated that the ability of the T4 motor to bypass inactive subunits "is fundamentally distinct from the strictly coordinated ϕ 29 and λ packaging motor." However, while the underlying mechanism may differ (it remains to be seen what the T4 mechanism is), the ϕ 29 motor has also been shown to possess the ability to bypass one inactive subunit by reassigning it as the non-translocating regulatory subunit (PMC3652982, PMC6077733). A more precise account of studies on other ring motors should be provided.

Reviewer #3 (Remarks to the Author):

Overall, this is a rigorous and interesting study describing motor stoichiometry and cooperativity amongst the subunits comprising the phage T4 DNA packaging motor. The presentation is dense and a bit difficult to wade through at times, but I have few suggestions on how to address given the amount of data is presented, which is significant. The biophysical approach represents a tour de force and demonstrates how sophisticated single molecule approaches can be utilized to dissect intimate details of motor function. Indeed, the real advance in this study is the ability to interrogate motors with a defined stoichiometry of inactive subunits and correlate this to activity using a variety of sophisticated fluorescence approaches. Importantly, the general approach can be employed in a number of biological systems, which will be of broad interest in the motor community. Details of my review are presented below, but in sum I believe that this manuscript is acceptable for publication in Nature Communications.

Main Points. In terms of mechanistic insight, two primary outcomes are evident.

First, photobleaching/TIRF studies indicate that 5 subunits assemble into a functional T4 motor. These are very nice experiment and the outcome clear, though pretty much anticipated. A bit of re-writing in this section would be useful. The motor is assembled with a 120 bp, unlabeled duplex and the complex should be described as an "assembled" motor rather than a "packaging" motor to avoid confusion. The later is not engendered until ATP is added in the next step at which time the Cy5 DNA substrate is also added. The order of presentation of the results is a bit confusing as written and it took a couple of reads of both Results and Methods to clearly define how the experiment was performed. Fig. 1a could be modified as well to clearly depict the experimental approach.

Also, the data presented in Fig. 1B shows 5 bleaching steps for the DNA substrate, which suggests that five substrates are packaged per pentameric motor, which is a bit misleading. This is addressed later in the manuscript but would be helpful to the reader to mention it here. Further, in all of the experiments, the unlabeled 120 bp "assembly" duplex must certainly be packaged, at least in part,

prior to binding and packaging of the labeled duplex. This is never discussed in the presentation of the data and really should be. How are the dwells in the tweezer studies affected by this?

Second, data is presented demonstrating that T4 motor is partially cooperative. Again, these are very nice experiments, the data are convincing and while unanticipated, not necessarily all that surprising; not all motors show strict cooperativity. Also, a couple of comments here. The bulk data presented in 2b clearly indicate that inactive subunits poison packaging activity and Fig. 2c are interpreted as demonstrating partial cooperativity; however, the fluorescence data presented in Fig. 2c shows zero fluorescent spots with a motor completely composed of mutant gp17. Since the slides were not apparently treated with DNase (?), does this mean that the mutant enzyme does not even bind DNA substrates? If so, then motor assembly with the unlabeled 120mer duplex must be affected as well? So, does the decrease in activity represent uncoordinated activity or defective motor assembly/stability? Was the Cy3 channel examined to confirm that motors were actually assembled in the presence of inactive motors? This is kinda, sorta addressed later (Fig. 3) but never directly addressed and it should be.

The study presented in Fig. 2 is really nice.

The optical trap data is also very nice and convincing; however, the preceding data indicates that motors with one and two inactive subunits are reasonably active. What is the likely proportion of these motors in the study, based on the binomial distribution analysis? How does this affect interpretation relative to the 3% presumption?

Notwithstanding, the tweezer data indicates that motors containing a defective subunit (at least one) spend more time in the quiescent state, which increases with the number of inactive subunits in the motor. Have the authors considered the simple hypothesis that the mutant subunit does not bind DNA at all (as suggested above) and that this decreases the probability that the pentameric motor binds DNA proportionally weaker? More significantly, while the T4 motor can clearly package DNA in the absence of the small terminase subunit, it is required for packaging *in vivo* and appears to be required for efficient initiation of packaging, i.e., the initial DNA binding event. This should be considered and discussed in the context of DNA binding events and the quiescent state, etc. Interestingly, supplemental Fig. 1e shows the effect of gp16 on ATPase activity of gp17 but the rationale for including this experiment is not discussed.

One final note. The presentation of the work is extremely "T4-centric". There are many other terminase motor systems that have been extensively characterized, kinetically, biochemically, biophysically and by single-molecule approaches, including phi29, lambda, P22, among others, but are given short shrift in the introduction and discussion of the data. Given the broad interests of Nature readership, an expanded presentation of how the present data fit into the field in general is warranted.

Minor points.

Page 3, second paragraph: While clear to me, the terms NTP and NTPase should be defined.

Page 4, second paragraph, “The fluorescence experimental design allowed us, for the first time, to precisely count the number of active/inactive motor subunits in each motor and observe packaging by individual motors in real-time.” As stated by the authors in the prior paragraph, a similar study has been performed in phi29 and this sentence should be modified to clarify that this type of study had not been performed in the T4 system.

Page 6, last paragraph: “Based purely on statistical ...”; it should be made clear that this section examines bulk packaging, rather than fluorescence data right off the bat to avoid confusion. Presentation of bulk and fluorescence data are both discussed prior to this. In fact, I would recommend moving Fig. 2e right after 2b to keep the bulk data together, and then move on to the fluorescence.

Fig. 2f. The panel shows that 5 duplexes are packaged and the presumption is that this is a fully wild-type motor, and thus no Cy3 data is presented. This should be clearly stated.

Fig. 3e: text states $f=0.27$ while figure states $f=0.26$ for ISD motors containing 2 defective subunits.

Page 10, second paragraph: A silly point, but the authors state “we referred to these motors as “arrested” because initial engagement of the DNA by the motor was successful but translocation was stopped”. In fact, the arrested motor was assembled but translocation never initiated and thus could not be stopped ...

In the Discussion (pg. 13, second paragraph), the state that the T4 motor alternates between a quiescent state which is free of DNA association and a DNA engaging state which is packaging competent. In fact, DNA is required to assemble the motor and it certainly must be present, bound to the motor at all times. This should be made clear.

Response to Reviewer Comments

The reviewer comments are in black, authors' responses are in blue, and how the comments are addressed in the revised manuscript are in yellow highlights.

1 Reviewer #1

1.1 In the manuscript by Dai et al., the authors present a series of single-molecule studies to examine the effect of mutant poisoning on the phage T4 DNA packaging motor. Most of the data is from well executed (and presumably very challenging) single particle fluorescence experiments and optical tweezers studies. The main conclusion is that the inclusion of one or more P-loop-mutant ATPases into the motor pentamer disrupts, but does not preclude, packaging initiation and DNA translocation. The authors conclude with a somewhat confusing interpretation that these results indicate that the motor is highly coordinated, but not so coordinated that it cannot overcome the presence of a defective subunit.

1.1.1 We thank the reviewer for the positive comments on the well-executed single-molecule studies presented in the paper and the constructive comments below.

1.2 Although well written, some of the text includes statements that seem to make inferences beyond the strength of the data and end with a mechanism that is self-contradicting. This manuscript will require significant revision to resolve these issues. Most of my criticisms are not what data is presented or analyzed, but how the results are interpreted and discussed relative to the existing literature.

1.2.1 We have clarified these concerns in the detailed responses given below and in the revised manuscript.

1.3 Abstract: Here and throughout the authors describe the ability of the motor to tolerate defective subunits as "unexpected". This is quite likely not the case as this is one of the few possible outcomes of such an experiment. This term should not be used.

1.3.1 We deleted "unexpectedly" from the abstract and other places in the manuscript.

1.4 Introduction: Line 90. Was the stoichiometry of the T4 motor in doubt? If not, the stoichiometry reported here is a reiteration, but not new.

1.4.1 As stated in the manuscript in lines 96-99, and 162-163, in the current study, we have determined the stoichiometry of the active phage T4 DNA packaging motor, which has not been done previously. Previous stoichiometry was determined by

cryo-EM reconstruction in which the motors were in a passive state, not engaged with DNA. Moreover, as mentioned in Introduction, a single-molecule study of phi29 determined a hexameric stoichiometry for both active and passive motor, in disagreement with structural studies (Shu *et al.*, 2007). So, the stoichiometry of the packaging motors remains a significant area of investigation and controversy.

- 1.5 Line 96. The statement “meaning that a single inactive subunit was sufficient to abolish packaging”, which is leaned on heavily throughout the manuscript, is incorrect. Whereas the bulk ensemble studies with Walker A mutant in lambda suggests a lack of tolerance for a defective subunit, in the paper by (Tafuya *et al.*, 2018). from the Bustamante group, motors with arginine-finger mutants were shown to package DNA with altered kinetics and stepping. Whereas the Tafuya paper does not quantify the number of mutant subunits like in this manuscript, it is with certainty that the motor can tolerate, in vitro, some defective subunits. This becomes an issue when the authors discuss their own results in attempting to draw a distinction between the mechanism and degree of coordination reported for other systems and that reported here for T4. Rather than diverging, to me these mechanisms appear to reinforcing each other, thus the central claim of T4 being somehow different seems inaccurate. The central claim that the data shows a dramatic difference between phage systems is not well supported.

- 1.5.1 We thank the reviewer for the above comments because we now realize that we have not fully explained this point in our manuscript. We do so now in the revised manuscript. In the case of phage λ motor, based on a statistical analysis of packaging activity in ensemble assays using a Walker A mutant in the large terminase subunit, it was concluded that the motor is strictly coordinated (100% coordination) (Andrews and Catalano, 2013). Thus, the presence of a single mutant subunit renders the motor inactive. In the case of phage phi29 motor, using a trans-arginine finger mutant in the terminase and single-molecule optical tweezers assays, it was found that one inactive subunit is tolerated (Tafuya *et al.*, 2018). However, such motors systematically pause every 10 bp in the dwell phase of each translocation cycle, resulting in a 10-fold reduction in the rate of packaging. Furthermore, motors containing two or more inactive subunits are completely inactive. In contrast, the behavior of the T4 DNA packaging motor in the presence of an inactive Walker A mutant subunit is distinct from either the phage λ motor or the phi29 motor. In ensemble assays, the motor shows partial coordination, unlike the phage λ motor. In single-molecule tweezers assays, unlike the phi29 motor, no periodic pausing was detected. Instead, random pauses and unpackaging events were observed (Fig. 4). In single-molecule fluorescence assays where we could count the number of inactive subunits in each motor, the T4 motor tolerated two or even three inactive subunits (Fig. 2). Furthermore, such ISD motors showed fewer DNA engagements. These behaviors have not been reported for the λ and phi29 motors. Hence, we conclude that the behavior of the T4 motor is distinct from that of the λ and phi29 motors.

- 1.5.2 The above points are included in the Discussion of the revised manuscript (lines 375-390 & 470-484). However, since the detailed mechanisms of coordination have not yet been established, stating "fundamentally distinct" may be too strong. Hence, in the revised manuscript, we simply state that the T4 motor's behavior is "distinct" from that of the λ and phi29 motors.
- 1.6 Results: Line 159. The Q163A mutant is described as impaired in hydrolysis. The authors should clarify whether or not, as a P-loop mutant, it is also defective in ATP binding since failure to bind ATP vs failure to hydrolyze have different implications when trying to interpret this phenotype in terms of what step is blocked by the mutant.
- 1.6.1 As described in the manuscript, the Q163A gp17 mutant is impaired in ATP hydrolysis. It retains motor assembly, forms hybrid motors with WT gp17 subunits, but shows complete loss of DNA packaging *in vitro*. Hence, it was used as an "inactive subunit" to analyze motor coordination. Our previously reported crystal structure (Sun *et al.*, 2007) shows that the Q163 residue is not in direct contact with the bound ATP, suggesting that the Q163A defect is most likely related to a post-ATP binding step, although the mechanism of impairment is unknown. This last point is added to Results (line 170-173) in the revised manuscript.
- 1.7 Lines 171-173 and 183. The assertion that any measurable activity in mutant doped rings can differentiate between a "strictly coordinated" and "not strictly coordinated" motor is contrary to how this has been interpreted in other phage packaging systems. The remainder of the paper is spent describing how mutant-doped motors differ in their behavior from wild type, indicative of the fact that all subunits participate in packaging and are thus coordinated in some way. Were the authors to discuss how their results show that coordination in T4 is the same as or different from other motors, and perhaps now better understood, the manuscript would add more to the ongoing discussion in this field.
- 1.7.1 The Discussion is revised by including additional details on different motors' behaviors and broadly discussing the similarities and distinctions between phi29 and T4 phage motors (lines 375-390 & 470-484).
- 1.8 Line 188. I believe this is meant to say "Q163A"
- 1.8.1 Corrected.
- 1.9 Lines 260-262. I'm not entirely sure how the wild type complexes are identified compared to the Cy3 spots. Are all Cy5 spots without Cy3 considered wild type? If so, for consistency are only Cy3 spots with DNA analyzed as part of the mutant-containing pool?
- 1.9.1 "Yes" to both questions.

1.10 Lines 280-281. If ATP-gamma-S allows the motor to bind but not package DNA, would one expect the same from the P-loop mutant? Again, the precise phenotype of the Q163A mutant would clarify this.

1.10.1 Yes, this expectation is likely, as evident from similar levels of transient DNA engagements by the WT and the ISD motors. However, the exact mechanism of the ATPase defect and its relation to DNA binding and/or readjustments by the motor are unknown and require a detailed investigation in the future.

1.11 Discussion: Lines 359-364. The argument that T4 is “fundamentally distinct” is not well supported. The authors invoke the idea that T4 is different because it is not “completely uncoordinated”. This is hair splitting, leaves the impression that the phage systems do not inform one another, and misses the opportunity to advance understanding of the packaging mechanism by reflecting on similarities rather than insisting they are different.

1.11.1 We have merely stated our findings. However, we agree that we should have provided more details on the coordination behaviors of different motors, which we have done in the revised manuscript in discussion section. As described above, in response to a similar comment, we have now included relevant details regarding the behaviors of T4 and other phage motors and toned down these statements (lines 375-390 & 470-484).

1.12 Lines 392-end. After having spent the entirety of the manuscript arguing that T4 is distinct from other studied packaging systems, the authors incorporate new structural studies on phi29 into their mechanism. What justifies this comparison? (although, as stated above, their results could be used to support an argument that these systems are indeed similar). The mechanism they proposed appears processive, ordinal, and therefore coordinated. Again, they argue that their mutant studies show that it doesn't have to be, but confusingly they have presented a mechanism that is coordinated. This is very confusing.

1.12.1 We respectfully disagree with the comment that we “*spent the entirety of the manuscript arguing that T4 is distinct from other studied packaging systems*”. On the contrary, most of our manuscript was devoted to analyzing the newly generated single-molecule fluorescence data of the T4 motor. We focused on this because this type of data is not available at present with any other phage motor system. In fact, one of the comments by Reviewer #3 was that the paper is too “T4-centric” and did not have enough discussion on other phage motors.

1.12.2 Regarding the reviewer's comment about the packaging mechanism, it is widely accepted that the phage and viral packaging motors and probably the ring

helicase motors exhibit common structural and functional features and translocate DNA by a common basic mechanism. We do believe this commonality to be the case and have stated so in the manuscript. Nevertheless, our findings suggest that some of the details of the mechanism, such as motor coordination, might have evolved differently, consistent with the respective phage life cycle.

- 1.12.3 Independently generated data from three different systems; i) our DNA gripping data from single-molecule tweezers studies suggesting that all five motor subunits might be involved in gripping DNA (Ordyan *et al.*, 2018), ii) structural data on phi29 packaging motor (Woodson *et al.*, 2021), and iii) atomic structures of ring helicase-DNA complexes (Yang *et al.*, 2019) and (Itsathitphaisarn *et al.*, 2012), all fit into a common translocation mechanism in which the motor assumes a helical shape as it grips the double-helical DNA. Therefore, under normal (WT motor) operation, the T4 motor, like the phi29 motor, is expected to fire in an ordinal manner, but when one or more inactive subunits are encountered, it has the flexibility to bypass them. This bypass is apparently because of the distinct behavior of the T4 motor coordination, as reported here.
- 1.12.4 This distinction might be because the phi29 motor uses a highly coordinated, dwell-burst, two-phase mechanism that is controlled at the motor level. In contrast, the T4 motor seems to have adapted a flexible mechanism that is controlled at the individual subunit level. Hence, the T4 motor is able to skip one, two, or even three inactive subunits by re-adjusting its DNA grip, whereas the phi29 motor re-sets the entire dwell-burst phase by making the ATPase-impaired subunit the “special non-translocating subunit”. Consequently, the duration of each dwell phase is greatly increased, resulting in a systematic long pause (dwell) in every translocation cycle, but the translocation (burst) phase is unaffected because the rest of the four motor subunits are functional and can hydrolyze ATP. However, the phi29 motor is not able to tolerate any more inactive subunits because both ATP binding and ATPase firing of a motor subunit are strictly dependent on coordinated activities of the previous subunit.
- 1.12.5 Our results, therefore, uncover the distinctive aspects of the T4 motor while still being consistent with an over-arching unified translocation mechanism for different phage motors.
- 1.12.6 We have revised the Discussion to bring out the above points more clearly (lines 375-390 & 470-484).

1.13 Line 440. The argument that the inability to abolish activity by modifying the motor “would confer survival advantages” suggests that the motor “evolved” to tolerate defects. Whereas I can understand that the motor could be seen to require flexibility to operate under a range of conditions, just because it can tolerate a mutant subunit doesn’t mean it evolved to be able to do so. If this were the case, where are the phages with defective ATPases?

1.13.1 We agree with the reviewer. In fact, in the Discussion, we have suggested that flexible T4 motor coordination leads to a faster packaging motor and reflects adaptation to package >8-times longer ~170-kb T4 DNA in the same amount of time as the shorter 19-kb DNA by phi29 motor. This flexibility also may have afforded the T4 motor to operate under a range of conditions (e.g., low ATP) but not necessarily because it has evolved to overcome defective motor subunits. As our data shows, though such motors can proceed with packaging, they do exhibit defective packaging dynamics and hence would not provide a survival advantage.

1.13.2 The statements in the Discussion are modified to make the above points clearer (lines 375-390 & 470-484).

2 Reviewer #2

2.1 In this work, Dai et al. investigated the intersubunit coordination of phage T4 DNA packaging motor using single-molecule TIRF and optical tweezers. By counting the number of inactive subunits in the ring motor via photobleaching and correlating it to the packaging behavior, the authors concluded that the motor consists of five subunits and can tolerate one inactive subunit, even though ISD impairs DNA engagement and translocation, arguing against a strictly coordination model. Overall, the data are of high quality and for the most part rigorously analyzed. The combination of multiple single-molecule techniques is a strength of the paper. However, I have the following concerns that should be addressed before the paper is acceptable for publication.

2.1.1 We thank the reviewer for the positive comments on the high quality of single-molecule data and analyses presented in the paper and the constructive comments below.

2.2 (1) In both Abstract and Introduction, the authors mentioned that they “precisely count the number of active/inactive motor subunits”. This is misleading. Because of <100% labeling efficiency, the number of inactive subunits was inferred by statistical probabilities, which was done for Fig 1c. However, this was not done for Fig 2g (one photobleaching step doesn't necessarily mean one inactive subunit, etc.). Can the authors take the labeling efficiency into consideration and reanalyze the data? If not, at the minimum this caveat should be clearly stated.

2.2.1 As suggested by the reviewer, we have added the following statement in lines 208-210 in the manuscript: “Due to <100% labeling efficiency and photoactivity, a small fraction of the photobleaching steps in the analysis shown in Figures 2-3, may represent >n inactive subunits.”

2.3 (2) In Fig 2e, why only 1- and 2-ISD data (red) are plotted? 0- and 3-ISD data are also measured from Fig 2f and 2g. Including all would make comparison with the bulk measurement (blue) and theoretical curves more conclusive.

2.3.1 We think the reviewer meant to say Fig. 3e, not Fig. 2e. Also, the reviewer seems to have missed the data for “0” inactive subunits in Fig. 3e.

2.3.2 We did not include the data for “3” inactive subunits because the trajectories are too few. Please note that the experiments shown in Fig. 2 involved steady-state imaging of multiple different areas within the reaction chamber, enabling a far greater number of single-molecule trajectories. However, for the experiments in Fig. 3e, observing the real-time interactions between the Cy5-labeled DNA and motor containing Cy3-labeled subunits required using a custom-built stopped-flow setup under the TIRF illumination. This setup allowed a single movie of a single and fixed area, thus restricting the number of single-molecule trajectories per experiment. Due to these limitations, the number of trajectories exhibiting three Cy3 photobleaching events, i.e., 3 inactive subunits, were low in number for the type of analysis shown in Fig. 3e and were thus not utilized for analysis.

2.4 (3) How did the authors distinguish between stable, semi-transient, and transient engagements? The distributions of their lifetimes inevitably overlap, especially for transient and semi-transient populations. A more detailed analysis should be conducted to show that the semi-transient population indeed exists.

2.4.1 To address this point, we include in the revised manuscript a distribution of DNA engagement lifetimes (Supplemental Fig. 8, also shown below). This distribution fits significantly better to a 3-exponential function than a 2-exponential function, showing that a model with 3 classes of engagements is the more appropriate one. The following statement is added to Results (lines 262-264): Distributions of

DNA engagement lifetimes were best fit by a 3-exponential model, validating our classification of DNA engagements into three characteristic types.

2.4.2
2.4.3

Supplementary Figure 8. Distributions of dwell times of DNA engagements.

(a) Survival probability distribution of the dwell times of the DNA engagements from a replicate experiment with the WT motor, and fits to a double- (blue) and triple-exponential functions (red). The inset shows representative dwell times of Cy5 signal, indicating DNA engagements, from a single-molecule fluorescence

trajectory. (b) Amplitudes and lifetimes from the triple-exponential fits to the distribution of dwell times of DNA engagements for motors with 0, 1, and 2 inactive subunits.

2.5 (4) The optical tweezers experiments showed heterogeneous velocities, pausing frequencies, and waiting times before the start. This is likely due to the different number of inactive subunits (one vs. two, for example) in the ring, which was not monitored in these experiments. The authors should comment on this issue and discuss why the heterogeneity in ISD motors is or is not higher than in WT motors.

2.5.1 As currently designed, the optical tweezers experiments do not provide a way of cross-validating the number of inactive subunits in ISD motors. In principle, cross-validation would be possible by fluorescently imaging dye-labeled inactive subunits while simultaneously monitoring the packaging of the motor complex with optical traps. However, such experiments are highly complex and require a specialized optical tweezers setup. Thus, while this is an area worthy of further investigation, it is beyond the scope of the current manuscript.

2.5.2 Regarding the heterogeneity, it is important to note that WT motors themselves are known to exhibit highly heterogeneous packaging dynamics (e.g., (Fuller *et al.*, 2007)), which reports high dynamic variability in T4 packaging). As a result, we cannot ascribe heterogeneous behavior to the number of inactive subunits in ISD motors. In our measurements, the standard deviation in packaging velocity represents 35% of the mean in WT motors and 40% in ISD motors. Thus the WT subunits in each type of motor could account for the majority of the variability in packaging speeds. The same argument is true for pause frequency and restart times, which are similarly variable in WT and ISD motors. As a result, differences in dynamics between WT and ISD motors can only be identified in the averages over the population of motors, as was done in our analysis.

2.5.3 We have clarified these points on the heterogeneity in packaging dynamics in Results, lines 334-340 in the revised manuscript.

2.6 (5) The amount of unpackaging seen in Fig. 4b is up to hundreds of base pairs, which seems at odds with the model depicted in Fig 5, where readjustment of the motor's DNA grip takes no more than a few bps.

2.6.1 Our data suggest that readjustment and unpackaging are related but distinct events. While the readjustment of the motor's DNA grip needs to occur in every translocation cycle in the case of an ISD motor and is too fast to resolve

individually in our assays, the unpacking events occur relatively rarely and often involve release of hundreds of basepairs of packaged DNA. We believe that unpacking represents events where the motor partially loses its grip on the DNA in the process of readjustment that, under the ~5 pN applied force, leads to loss of DNA registry with motor and DNA release. When the DNA grip and registry is restored, the motor is reset and packaging continues. As stated in Results and Discussion (lines 354-356 & 462-467), this type of unpacking behavior has also been observed previously under conditions of low ATP concentration when a subunit is not occupied with ATP (Kottadiel *et al.*, 2012), a situation similar to the presence of an ATPase-defective subunit described in this study.

2.7 (6) The authors stated that the ability of the T4 motor to bypass inactive subunits “is fundamentally distinct from the strictly coordinated ϕ 29 and λ packaging motor.” However, while the underlying mechanism may differ (it remains to be seen what the T4 mechanism is), the ϕ 29 motor has also been shown to possess the ability to bypass one inactive subunit by reassigning it as the non-translocating regulatory subunit (PMC3652982, PMC6077733). A more precise account of studies on other ring motors should be provided.

2.7.1 We agree with the reviewer’s comment, and a similar comment was also made by reviewer #1. In the case of phage λ motor, based on statistical analysis of packaging activity in ensemble assays using a Walker A mutant in the large terminase subunit, it was concluded that the motor is strictly coordinated (100% coordination) (Andrews and Catalano, 2013). Thus, the presence of a single mutant subunit renders the motor inactive. In the case of phage phi29 motor, using a trans-arginine finger mutant in the terminase and single-molecule optical tweezers assays, it was found that one inactive subunit is tolerated (Tafuya *et al.*, 2018). However, such motors systematically pause every 10 bp in the dwell phase of each translocation cycle, resulting in a 10-fold reduction in the rate of packaging. Furthermore, motors containing two or more inactive subunits are completely inactive.

2.7.2 In contrast, the behavior of the T4 DNA packaging motor in the presence of an inactive Walker A mutant subunit is distinct from either the phage λ motor or the phi29 motor. In ensemble assays, the motor shows partial coordination, unlike the phage λ motor. In single molecule tweezers assays, unlike the phi29 motor, no periodic pausing every 10 bp was detected. Instead, random pauses and unpacking events were observed (Fig. 4). In single-molecule fluorescence assays where we could count the number of inactive subunits in each motor, the T4 motor tolerated two, or even three inactive subunits (Fig. 2). Furthermore, such ISD motors showed fewer DNA engagements. These behaviors have not been reported for the λ and phi29 motors. Hence, we conclude that the behavior of the T4 motor is distinct from that of the λ and phi29 motors.

- 2.7.3 The above points are included in the Discussion of the revised manuscript (lines 375-390 & 470-484). Since the detailed mechanisms of coordination have not yet been established, stating “fundamentally distinct” may be too strong. Hence, in the revised manuscript, we simply state that the T4 motor’s behavior is “distinct” from that of the λ and phi29 motors.

3 Reviewer #3

- 3.1 Overall, this is a rigorous and interesting study describing motor stoichiometry and cooperativity amongst the subunits comprising the phage T4 DNA packaging motor. The presentation is dense and a bit difficult to wade through at times, but I have few suggestions on how to address given the amount of data is presented, which is significant. The biophysical approach represents a tour de force and demonstrates how sophisticated single-molecule approaches can be utilized to dissect intimate details of motor function. Indeed, the real advance in this study is the ability to interrogate motors with a defined stoichiometry of inactive subunits and correlate this to activity using a variety of sophisticated fluorescence approaches. Importantly, the general approach can be employed in a number of biological systems, which will be of broad interest in the motor community. Details of my review are presented below, but in sum I believe that this manuscript is acceptable for publication in Nature Communications.

- 3.1.1 We thank the reviewer for the positive comments on several aspects of our single-molecule analyses presented in the paper and the constructive comments below.

- 3.2 Main Points. In terms of mechanistic insight, two primary outcomes are evident. First, photobleaching/TIRF studies indicate that 5 subunits assemble into a functional T4 motor. These are very nice experiment and the outcome clear, though pretty much anticipated. A bit of re-writing in this section would be useful. The motor is assembled with a 120 bp, unlabeled duplex and the complex should be described as an “assembled” motor rather than a “packaging” motor to avoid confusion. The later is not engendered until ATP is added in the next step at which time the Cy5 DNA substrate is also added. The order of presentation of the results is a bit confusing as written and it took a couple of reads of both Results and Methods to clearly define how the experiment was performed. Fig. 1a could be modified as well to clearly depict the experimental approach.

- 3.2.1 As per the reviewer’s suggestion, the term “assembled motor” was used instead of “packaging motor” to describe the experimental setup in Materials and Methods. A Supplementary Figure 2a (also shown below) containing a schematic depicting all the details of the experimental setup used for the single-molecule fluorescence experiments is included in the revised manuscript. Additional details on the steps used for the assembly of the head-motor complexes are included in the first section of Results and in Methods in the revised manuscript (lines 129-134 & 764-771).

3.2.2
3.2.3

Supplementary Figure 2. Controls showing that all entities of an immobilized packaging motor are required to obtain Cy3 and Cy5 signals in single-molecule assays, validating these assays' authenticity. (a) Schematic depicting the steps involved in the reconstitution of the assembled T4 motor for single-molecule fluorescence studies. Homomeric gp17 subunits constituting the motor are labeled with Cy3. (b) Individual spots in the Cy3 channel (right) are indicative of surface-immobilized Cy3 labeled motor. Individual spots in the Cy5 channel (left) are indicative of the DNA packaged into the capsid by the surface-immobilized Cy3 labeled motor (middle panel). In the control panel (left panel) lacking capsid and motor, rare non-specific background spots are seen. In another control lacking the motor, only a few spots are seen due to the association of DNA to the capsid portal in the absence of the motor (right panel).

3.3 Also, the data presented in Fig. 1B shows 5 bleaching steps for the DNA substrate, which suggests that five substrates are packaged per pentameric motor, which is a bit misleading. This is addressed later in the manuscript but would be helpful to the reader to mention it here.

3.3.1 We would like to make a friendly correction for the reviewer comment, the representative trajectory shown in Fig. 1b shows 4 Cy5 photobleaching steps, not 5. For the experiments shown in Figure 1, the Cy5 signal was merely used to ascertain the packaging capability of the single-motor complexes being analyzed. The number of Cy5 photobleaching steps were not required and thus not used in

the analysis for the data shown in Figure 1. In response to the reviewer's comment, we have added the following statement in the legend of Figure 1: "The number of Cy5 photobleaching steps, indicating the number of packaged DNA, can range from 1 to many"

3.4 Further, in all of the experiments, the unlabeled 120 bp "assembly" duplex must certainly be packaged, at least in part, prior to binding and packaging of the labeled duplex. This is never discussed in the presentation of the data and really should be. How are the dwells in the tweezer studies affected by this?

3.4.1 Based on optimization experiments performed previously (Fuller *et al.*, 2007); (Vafabakhsh *et al.*, 2014)), it was observed that the 120-bp DNA (and ATP- γ S) enhances the efficiency of assembly of functional packaging complexes in single-molecule studies. This DNA is probably bound to the motor but not translocated. However, it is possible that some of these molecules get translocated soon after ATP is added, before the first Cy5 labeled DNA molecule is engaged with the motor. However, please note that, in the current optical tweezers experiments, we did not need to use the 120-bp DNA because the packaging complexes were assembled by replacing the 120-bp DNA with the 3.4-kb DNA substrate. As per the reviewer's suggestion, the use of 120-bp DNA is clarified in Results and Methods in the revised manuscript (lines 129-134, 764-771, & 846-849).

3.5 Second, data is presented demonstrating that T4 motor is partially cooperative. Again, these are very nice experiments, the data are convincing and while unanticipated, not necessarily all that surprising; not all motors show strict cooperativity. Also, a couple of comments here. The bulk data presented in 2b clearly indicate that inactive subunits poison packaging activity and Fig. 2c are interpreted as demonstrating partial cooperativity; however, the fluorescence data presented in Fig. 2c shows zero fluorescent spots with a motor completely composed of mutant gp17. Since the slides were not apparently treated with DNase (?), does this mean that the mutant enzyme does not even bind DNA substrates?

3.5.1 The reviewer seems to have misread the protocol because the data shown in Fig. 2d (not 2c) were after DNase I treatment. Therefore, any non-encapsidated DNA molecules would have been removed from the chamber. Hence, there would be no Cy5 spots if there was no packaging.

3.6 If so, then motor assembly with the unlabeled 120mer duplex must be affected as well? So, does the decrease in activity represent uncoordinated activity or defective motor assembly/stability? Was the Cy3 channel examined to confirm that motors were actually assembled in the presence of inactive motors? This is kinda, sorta addressed later (Fig. 3) but never directly addressed and it should be.

- 3.6.1 Motors containing 1-3 inactive subunits, show DNA engagements of all kinds, similar in quality to those seen with the WT motor. These motors with inactive subunits also showed consistent packaging activity in optical trap experiments indicating that inactive subunits are not impaired in motor assembly.
- 3.6.2 Moreover, we conducted a series of experiments to determine the stoichiometry of motors consisting solely of Cy3-labeled Q163A mutant and confirmed that not only does the Q163A mutant subunits bind to heads but also that the stoichiometry of the mutant motor is a pentamer, same as the WT motor. We have included these additional data in Supplementary Fig. 3b-d. Furthermore, we performed another additional analysis showing that the propensity of the inactive subunit assembling into a pentameric motor is nearly the same as the WT motor (Supplementary Fig. 3e-g). The above data clearly demonstrate that the mutant subunits are not defective in motor assembly/stability.
- 3.6.3 A Supplementary figure on the stoichiometry of Cy3-labeled Q163A gp17 mutant is included in the revised manuscript and a sentence stating that the Q163A mutant retains motor assembly is added to Results (Supplementary Fig. 3b-g (also shown below), lines 172-173).

3.6.4
3.6.5

Supplementary Figure 3. Assessing the stoichiometry of the WT motor and propensity of inactive subunits in proper motor assembly. (a) Quality of fits to the distribution of Cy3 photobleaching steps assuming different putative stoichiometries of the motor. (b-d) Inactive subunits also assemble to form pentameric motor. (b) Schematic of assay similar to that in Figure 1a, but using Cy3 labeled inactive subunits only. (c) Representative single-molecule fluorescence trajectories, showing different number of Cy3 photobleaching steps. (d) Distribution of the number Cy3 photobleaching steps pooled from multiple trajectories. With the Cy3 labeling efficiency of the inactive subunit at 68%, the best binomial fit (blue curve) yielded the stoichiometry of 5, confirming that inactive subunits can assemble into a pentameric motor complex, same as WT motor subunits. (e-g) Inactive subunits and WT subunits have the same propensity of assembly into the pentameric motor. (e) Schematic of assay similar that in Figure 1a, but using motors doped with 20% of Cy3-labeled inactive subunits, as in the experiments described in Fig. 2f and Fig. 3. (f) Representative single-molecule fluorescence trajectories, showing different number of Cy3

photobleaching steps. (g) Distribution of the number of Cy3 photobleaching steps pooled from multiple trajectories and fit. The binomial fit to the distribution gives the doping % of 21%, close to the experimental value of 20%, showing that inactive subunits have the same propensity as WT subunits to assemble into pentameric motors.

3.7 The study presented in Fig. 2 is really nice. The optical trap data is also very nice and convincing; however, the preceding data indicates that motors with one and two inactive subunits are reasonably active. What is the likely proportion of these motors in the study, based on the binomial distribution analysis? How does this affect interpretation relative to the 3% presumption?

3.7.1 Based on the binomial distribution, we expect 47% of ISD motors to contain 1 or 2 inactive subunits and only 3% of the motors to contain 0 inactive subunits. In the optical tweezers experiment, the probability of observing activity in ISD motors was found to be about 50% that for WT motors, which is consistent with the presumption that motors containing 0, 1, or 2 inactive subunits (3 + 47%) can package DNA. We have included this point in the revised manuscript on lines 307-309 & 316-318.

3.8 Notwithstanding, the tweezer data indicates that motors containing a defective subunit (at least one) spend more time in the quiescent state, which increases with the number of inactive subunits in the motor. Have the authors considered the simple hypothesis that the mutant subunit does not bind DNA at all (as suggested above) and that this decreases the probability that the pentameric motor binds DNA proportionally weaker

3.8.1 It is unlikely that the basic DNA binding function is lost in the Q163A mutant, as shown by similar levels of transient DNA engagements between the WT and the ISD motors. However, the subsequent steps leading to a more stable interaction and encapsidation (semi-transient and long-lived engagements) are defective in the ISD motors.

3.9 More significantly, while the T4 motor can clearly package DNA in the absence of the small terminase subunit, it is required for packaging *in vivo* and appears to be required for efficient initiation of packaging, i.e., the initial DNA binding event. This should be considered and discussed in the context of DNA binding events and the quiescent state, etc.

3.9.1 While gp16 is essential for packaging initiation *in vivo*, this effect could only be recapitulated *in vitro* in crude packaging systems for packaging the concatemeric DNA (Leffers and Rao, 2000). In fact, gp16 has an opposite effect, inhibition of *in vitro* DNA packaging, in the defined packaging system reconstituted with purified heads, motor protein gp17, and linear plasmid DNA as substrate. Therefore,

while we cannot rule out the potential effect of gp16 on the quiescent state, it is difficult to make a connection at this stage without more evidence.

3.10 Interestingly, supplemental Fig. 1e shows the effect of gp16 on ATPase activity of gp17, but the rationale for including this experiment is not discussed.

3.10.1 Supplemental Fig. 1e is included to demonstrate that the Q163A mutant is defective in gp17-ATPase activity, which is stimulated in the presence of gp16 (in the absence of DNA packaging) ((Leffers and Rao, 2000); (Kondabagil *et al.*, 2006)). The above point is added to the figure legend of Supplementary Figure 1.

3.11 One final note. The presentation of the work is extremely “T4-centric”. There are many other terminase motor systems that have been extensively characterized, kinetically, biochemically, biophysically and by single-molecule approaches, including phi29, lambda, P22, among others, but are given short shrift in the introduction and Discussion of the data. Given the broad interests of Nature readership, an expanded presentation of how the present data fit into the field in general is warranted.

3.11.1 We appreciate this comment and in fact struggled with it ourselves. Because there were no other reports on counting the inactive subunits of single T4 packaging motors, we focused on this distinctive aspect of our studies and did not want to distract the reader with general discussions. To address this and related comments by other reviewers, we have included information on phage packaging motors and discussion on their coordination behaviors in Introduction and Discussion sections of the revised manuscript, and appropriate references are also added (lines 84-93, 375-390 & 470-484).

References:

- Andrews, BT, and Catalano, CE (2013). Strong subunit coordination drives a powerful viral DNA packaging motor. *Proc Natl Acad Sci U S A* 110, 5909–5914.
- Fuller, DN, Raymer, DM, Kottadiel, VI, Rao, VB, and Smith, DE (2007). Single phage T4 DNA packaging motors exhibit large force generation, high velocity, and dynamic variability. *Proc Natl Acad Sci U S A* 104, 16868–16873.
- Itsathitphaisarn, O, Wing, RA, Eliason, WK, Wang, J, and Steitz, TA (2012). The hexameric helicase DnaB adopts a nonplanar conformation during translocation. *Cell* 151, 267–277.
- Kondabagil, KR, Zhang, Z, and Rao, VB (2006). The DNA translocating ATPase of bacteriophage T4 packaging motor. *J Mol Biol* 363, 786–799.
- Kottadiel, VI, Rao, VB, and Chemla, YR (2012). The dynamic pause-unpackaging state, an off-translocation recovery state of a DNA packaging motor from bacteriophage T4. *Proc Natl Acad Sci U S A* 109, 20000–20005.
- Leffers, G, and Rao, VB (2000). Biochemical characterization of an ATPase activity associated with the large packaging subunit gp17 from bacteriophage T4. *J Biol Chem* 275, 37127–37136.
- Ordyan, M, Alam, I, Mahalingam, M, Rao, VB, and Smith, DE (2018). Nucleotide-dependent DNA gripping and an end-clamp mechanism regulate the bacteriophage T4 viral packaging motor. *Nat Commun* 9, 5434.

Shu, D, Zhang, H, Jin, J, and Guo, P (2007). Counting of six pRNAs of phi29 DNA-packaging motor with customized single-molecule dual-view system. *EMBO J* 26, 527–537.

Sun, S, Kondabagil, K, Gentz, PM, Rossmann, MG, and Rao, VB (2007). The structure of the ATPase that powers DNA packaging into bacteriophage T4 procapsids. *Mol Cell* 25, 943–949.

Tafoya, S, Liu, S, Castillo, JP, Atz, R, Morais, MC, Grimes, S, Jardine, PJ, and Bustamante, C (2018). Molecular switch-like regulation enables global subunit coordination in a viral ring ATPase. *Proc Natl Acad Sci U S A* 115, 7961–7966.

Vafabakhsh, R, Kondabagil, K, Earnest, T, Lee, KS, Zhang, Z, Dai, L, Dahmen, KA, Rao, VB, and Ha, T (2014). Single-molecule packaging initiation in real time by a viral DNA packaging machine from bacteriophage T4. *Proc Natl Acad Sci U S A* 111, 15096–15101.

Woodson, M, Pajak, J, Mahler, BP, Zhao, W, Zhang, W, Arya, G, White, MA, Jardine, PJ, and Morais, MC (2021). A viral genome packaging motor transitions between cyclic and helical symmetry to translocate dsDNA. *Sci Adv* 7.

Yang, L et al. (2019). Terminase Large Subunit Provides a New Drug Target for Herpesvirus Treatment. *Viruses* 11.

REVIEWER COMMENTS

Reviewer #1 (Remarks to the Author):

I greatly appreciate the time and attention taken by the authors to address the concerns of the reviewers. Much of the changes in the text have clarified or addressed the reviewers' comments.

The main point that I am still struggling to accept is how the authors go from the observation that Walker A mutants are tolerated by the motor that forms a helical ring that processively moves the DNA through the motor channel as illustrated in Fig. 5. They offer no direct evidence of the cracked ring mechanism described, and there is considerable hand waving about how their data supports such a mechanism given that the model presented clearly suggests a high degree of coordination and processivity, which the toleration of mutants argues against. The authors state "Figure 5b-c shows a model of DNA translocation based on our data." This is simply not the case.

Reviewer #2 (Remarks to the Author):

The authors did a good job addressing my critiques in the revised manuscript. I have one additional comment on Fig. 2e. The blue and red curves measured different things: the blue curve measures the total amount of DNA packaged, which is determined by both the fraction of active particles and the packaging activity of each particle; while the red curve only measures the fraction of active particles. In order to directly compare them, the Cy5 intensity at each spot should be taken into consideration as the authors did in 2f and 2g. For this reason I suggest that the two curves not be overlaid.

Otherwise I think the manuscript is acceptable for publication.

Reviewer #3 (Remarks to the Author):

The authors have adequately responded to the majority of my comments and concerns, though one suggestion and a couple of comments remain (see below). The study remains a tour-de-force of single molecule approaches that rigorously interrogate T4 motor assembly and function. As stated in my first review, it is in my opinion suitable for publication in Nature.

One remaining issue; I previously suggested that a discussion of the TerS subunit in T4 packaging should be considered and this suggestion stands. In their response, the authors confirm that TerS is strictly required in vivo and stimulates packaging of wild-type concatemeric DNA substrates in vitro. In contrast, TerS has been shown to inhibit packaging when short, linear packaging substrates are used in vitro. This has been and remains a conundrum that has never really been address, as far as I know. Data presented in this study may provide a clue.

The results show that most (~80%) DNA engagements in the active state fail to lead to encapsidation regardless of motor composition (Lines 415-417; Fig. 3d, left panel). In other words, packaging initiation is poor with both WT and ISD motors. The authors state that “these transient events must correspond to DNA coming into loose contact with the motor and then dissociating (Lines 415-417). The authors speculate that the use of short duplexes may be responsible for poor initiation. It is also possible that TerS is required for efficient motor assembly and/or efficient transition to the translocating motor. In their response, the authors state “while we cannot rule out the potential effect of gp16 on the quiescent state, it is difficult to make a connection at this stage without more evidence”. While true, the authors speculate about a variety of details of motor function and this would be a perfect place to consider the requisite role of TerS in T4 genome packaging since the data provides potential insight.

Something to consider; the packaging motor model while speculative is reasonable and the authors speculate that the partially uncoordinated nature of the T4 motor may play a biological role in packaging the long, concatemeric T4 genome, which is also reasonable. An additional feature of T4 that this may address is the highly branched structure of newly replicated concatemeric T4 DNA, which may similarly benefit from a more plastic motor. Just a thought.

One last quibble; the authors describe ATP- γ S as a “non-hydrolyzable” ATP analog. This is not correct as many ATPases can hydrolyze this nucleotide analog quite efficiently, including lambda terminase. Unless this has been directly demonstrated for T4 terminase, it is more appropriately described as a “poorly” or “slowly” hydrolyzed ATP analog.

RESPONSES TO REVIEWERS' COMMENTS

Please note that most of the reviewers' comments below are new, not part of the original reviews. However, we are pleased to address these comments in the current revised manuscript.

Reviewer #1 (Remarks to the Author):

I greatly appreciate the time and attention taken by the authors to address the concerns of the reviewers. Much of the changes in the text have clarified or addressed the reviewers' comments.

We thank the reviewer for the positive comments.

The main point that I am still struggling to accept is how the authors go from the observation that Walker A mutants are tolerated by the motor that forms a helical ring that processively moves the DNA through the motor channel as illustrated in Fig. 5. They offer no direct evidence of the cracked ring mechanism described, and there is considerable hand waving about how their data supports such a mechanism given that the model presented clearly suggests a high degree of coordination and processivity, which the toleration of mutants argues against. The authors state "Figure 5b-c shows a model of DNA translocation based on our data." This is simply not the case.

Please note that our model is not a "cracked ring mechanism" as mentioned by the reviewer.

As described in our manuscript, we proposed a model based on our data as well as the available information in the literature. The term "based on our data" refers to our single molecule data that provide the basis for proposing a "flexible" packaging model for the T4 motor. However, we can state this better. Therefore, we deleted "based on our data" from the sentence and slightly modified the sentence such that it also refers to several considerations described in the previous paragraph that led to the proposed model.

Reviewer #2 (Remarks to the Author):

The authors did a good job addressing my critiques in the revised manuscript. I have one additional comment on Fig. 2e. The blue and red curves measured different things: the blue curve measures the total amount of DNA packaged, which is determined by both the fraction of active particles and the packaging activity of each particle; while the red curve only measures the fraction of active particles. In order to directly compare them, the Cy5 intensity at each spot should be taken into consideration as the authors did in 2f and 2g. For this reason I suggest that the two curves not be overlaid.

We thank the reviewer for the positive comments.

Fig. 2 is modified as suggested.

Otherwise I think the manuscript is acceptable for publication.

Reviewer #3 (Remarks to the Author):

The authors have adequately responded to the majority of my comments and concerns, though one suggestion and a couple of comments remain (see below). The study remains a tour-de-force of single molecule approaches that rigorously interrogate T4 motor assembly and function. As stated in my first review, it is in my opinion suitable for publication in Nature.

We thank the reviewer for the positive comments.

One remaining issue; I previously suggested that a discussion of the TerS subunit in T4 packaging should be considered and this suggestion stands. In their response, the authors confirm that TerS is strictly required in vivo and stimulates packaging of wild-type concatemeric DNA substrates in vitro. In contrast, TerS has been shown to inhibit packaging when short, linear packaging substrates are used in vitro. This has been and remains a conundrum that has never really been address, as far as I know. Data presented in this study may provide a clue.

The results show that most (~80%) DNA engagements in the active state fail to lead to encapsidation regardless of motor composition (Lines 415-417; Fig. 3d, left panel). In other words, packaging initiation is poor with both WT and ISD motors. The authors state that “these transient events must correspond to DNA coming into loose contact with the motor and then dissociating (Lines 415-417). The authors speculate that the use of short duplexes may be responsible for poor initiation. It is also possible that TerS is required for efficient motor assembly and/or efficient transition to the translocating motor. In their response, the authors state “while we cannot rule out the potential effect of gp16 on the quiescent state, it is difficult to make a connection at this stage without more evidence”. While true, the authors speculate about a variety of details of motor function and this would be a perfect place to consider the requisite role of TerS in T4 genome packaging since the data provides potential insight.

As suggested, a short paragraph discussing the potential role of the small terminase is included in Discussion.

Something to consider; the packaging motor model while speculative is reasonable and the authors speculate that the partially uncoordinated nature of the T4 motor may play a biological role in packaging the long, concatemeric T4 genome, which is also reasonable. An additional feature of T4 that this may address is the highly branched structure of newly replicated concatemeric T4 DNA, which may similarly benefit from a more plastic motor. Just a thought.

As suggested, a statement regarding the potential advantage of a flexible motor to package concatemeric and branched T4 DNA is included in Discussion.

One last quibble; the authors describe ATP- γ S as a “non-hydrolyzable” ATP analog. This is not correct as many ATPases can hydrolyze this nucleotide analog quite efficiently, including lambda terminase. Unless this has been directly demonstrated for T4 terminase, it is more appropriately described as a “poorly” or “slowly” hydrolyzed ATP analog.

As suggested, the term “non-hydrolyzable” ATP analog is changed to “poorly hydrolyzed” ATP analog at several places in the text.

REVIEWERS' COMMENTS

Reviewer #3 (Remarks to the Author):

From the start this was a great body of work. All of my comments have been addressed and I believe the manuscript is suitable for publication in Nature